# Whole genome and transcriptome integrated analyses guide clinical care of pediatric poor prognosis cancers

Rebecca J. Deyell [1,8] ✉, Yaoqing Shen[2,8], Emma Titmuss [2,8], Katherine Dixon[2,3], Laura M. Williamson[2], Erin Pleasance [2], Jessica M. T. Nelson[2], Sanna Abbasi[2], Martin Krzywinski[2], Linlea Armstrong[3], Melika Bonakdar [2], Carolyn Ch'ng [2], Eric Chuah[2], Chris Dunham[4], Alexandra Fok[2], Martin Jones [2], Anna F. Lee [4], Yussanne Ma[2], Richard A. Moore[2], Andrew J. Mungall [2], Karen L. Mungall[2], Paul C. Rogers[1], Kasmintan A. Schrader [3], Alice Virani[3], Kathleen Wee[2], Sean S. Young [3,5], Yongjun Zhao[2], Steven J. M. Jones [2,3,6,9], Janessa Laskin [7,9], Marco A. Marra [2,3,9] & Shahrad R. Rassekh [1,9] ✉

The role for routine whole genome and transcriptome analysis (WGTA) for poor prognosis pediatric cancers remains undetermined. Here, we characterize somatic mutations, structural rearrangements, copy number variants, gene expression, immuno-profiles and germline cancer predisposition variants in children and adolescents with relapsed, refractory or poor prognosis malignancies who underwent somatic WGTA and matched germline sequencing. Seventy-nine participants with a median age at enrollment of 8.8 y (range 6 months to 21.2 y) are included. Germline pathogenic/likely pathogenic variants are identified in 12% of participants, of which 60% were not known prior. Therapeutically actionable variants are identified by targeted gene report and whole genome in 32% and 62% of participants, respectively, and increase to 96% after integrating transcriptome analyses. Thirty-two molecularly informed therapies are pursued in 28 participants with 54% achieving a clinical benefit rate; objective response or stable disease ≥6 months. Integrated WGTA identifies therapeutically actionable variants in almost all tumors and are directly translatable to clinical care of children with poor prognosis cancers.

Despite modern approaches to therapy, an unacceptably high proportion of children and adolescents diagnosed with cancer continue to fail upfront therapy[1]. Advances in our understanding of somatic genomic alterations in pediatric cancers have enhanced risk stratification schema and facilitated the development of a limited number of effective targeted therapeutic agents[2–5]. Despite this, at relapse or disease progression, pediatric cancers have almost uniformly dismal survival outcomes across cancer types, and second-line

[1]Department of Pediatrics, BC Children's Hospital and Research Institute, Vancouver, BC, Canada. [2]Canada's Michael Smith Genome Sciences Centre at BC Cancer, Vancouver, BC, Canada. [3]Department of Medical Genetics, University of British Columbia, Vancouver, BC, Canada. [4]Department of Pathology and Laboratory Medicine, University of British Columbia, Vancouver, BC, Canada. [5]Cancer Genetics and Genomics Laboratory, Department of Pathology and Laboratory Medicine, BC Cancer, Vancouver, Canada. [6]Department of Molecular Biology and Biochemistry, Simon Fraser University, Burnaby, BC, Canada. [7]Department of Medical Oncology, BC Cancer, Vancouver, BC, Canada. [8]These authors contributed equally: Rebecca J. Deyell, Yaoqing Shen, Emma Titmuss. [9]These authors jointly supervised this work: Steven J. M. Jones, Janessa Laskin, Marco A. Marra, Shahrad R. Rassekh. ✉e-mail: rdeyell@cw.bc.ca; rrassekh@cw.bc.ca

therapies remain predominantly untargeted with significant toxicity and minimal efficacy[6,7].

Recognition of the stagnating reduction of childhood cancer mortality rates[8], coupled with increasingly accessible and sophisticated next generation sequencing (NGS) and bioinformatics, has fueled a new age of molecular discovery in pediatric oncology. Access to comprehensive genomic analyses has rapidly accelerated with the establishment of multiple, large pediatric cancer precision medicine programs[9–18]. These trials demonstrate the utility of NGS assays in multiple categories of actionability including refining diagnoses, discovery of underlying cancer predisposition in 7–18% of participants and identifying therapeutically actionable variants in up to 70–86%, with variable level of evidence (LOE) schema. Many patients have more than one potentially actionable therapeutic variant, and this becomes increasingly complex as we attempt to incorporate expression profiles into therapeutic decision making[10,12,16,18,19].

A unified genomics and bioinformatics pipeline, the Personalized OncoGenomics (POG) program, was developed to provide access to whole genome and transcriptome analysis (WGTA) for poor prognosis cancer patients of all ages and diagnoses in British Columbia (BC), Canada in 2012[20]. This program was initiated with a mandate to provide timely access to WGTA for advanced cancer patients and has now enrolled >1200 participants[21,22]. In this work, we describe the feasibility, utility, and clinical benefit of the pediatric POG program since its inception, with a focus on the clinical actionability of identified therapeutic targets.

## Results

### Pediatric patient cohort characteristics and molecular tumor board timing

Ninety-one patients were approached and offered enrollment into the pediatric POG study between September 2013 and July 2019, with 88 (96.7%) consenting to participate in the study (Fig. 1A). One hundred tumor samples from 87 participants were submitted for sequencing analysis. In total, 89.7% of participants (n = 78) received results from a completed triad of somatic WGS, somatic RNA, and germline WGS sequencing, and one additional patient received only WGS results due to RNA-library failure (n = 79 in analytic cohort, Source Data file). One patient was concurrently diagnosed with two cancers (central nervous system (CNS) atypical teratoid rhabdoid tumor and renal rhabdoid tumor) and both cancers were sequenced, while three other patients were successfully sequenced twice for a total of 83 somatic WGTA samples (Fig. 1A). Nine enrolled participants were excluded from the study cohort due to low tumor content or low DNA yield (n = 8 participants) or no biopsy (n = 1) (Supplementary Table S1).

In the final study cohort (n = 79), 43% of the participants were female and the median age at cancer diagnosis and study enrollment was 8.8 y (range 0–20.7 y) and 13.4 y (range 0.5–21.2 y), respectively (Fig. 1B). The cohort included 49 (59%) solid, non-CNS, 19 (23%) CNS, 9 (11%) benign, and 6 (7%) hematologic malignancies (Table 1). The most common tumor types were soft tissue and bone sarcoma (n = 25), high grade primary CNS tumors (n = 16), and neuroblastoma (n = 9), with samples obtained from a range of biopsy sites (Fig. 1C). Locally invasive, unresectable benign tumors such as aggressive fibromatosis and plexiform neurofibromas were also included (n = 9). Biopsies were obtained from the primary disease site in 63% of cases and 70% of samples were obtained at the time of a clinically indicated surgery or biopsy. Thirty-three percent (n = 27) of tumor samples were collected and submitted for WGTA within one month of initial diagnosis (Table 1). The 2-year event-free survival (EFS) for the study cohort from the time of enrollment was 28.6 ± 10.0% with a median time to progression of 6.1 months, which did not vary significantly by tumor category (Fig. 1D; P = 0.083). The 2-year overall survival (OS) was 48.5 ± 11.2% and varied significantly by tumor group (Fig. 1E; P = 0.006), with no deaths among those with benign tumors. The median follow-

up for cohort participants from the time of first study sample receipt for sequencing was 21.8 (range 0.1–78.3) months.

Return of sequencing results was stepwise, with an expedited initial disclosure of the rapid targeted gene report (TGR) at a median of 34 days from sample receipt (Table 1). The full WGTA report, including a pathway diagram and table of actionable variants annotated by LOE, was released prior to full molecular tumor board (MTB) meetings. MTBs were held at a median 71 days from sample receipt, with 24% presented between 30 and 60 days. Of note, 19 participants died prior to or within 3 months of their MTB presentation date, representing 24% of participants with sequencing results.

### Characterization of participant germline findings

Germline variant prioritization and interpretation was guided by advances in biological and clinical knowledge and evolution of standardized variant classification guidelines (Fig. 2A)[23]. During the study, 91 germline findings identified in 46 participants (57%) were reviewed for potential clinical actionability (Fig. 2B). Curation for *PMS2* and *MUTYH* variants were among the most common. At the time of prospective analysis, eight pathogenic (P) or likely pathogenic (LP) variants were reviewed, of which five were known prior to POG. Pathogenic variants in *PTEN* and *CHEK2* identified in two children were returned to treating oncologists who disclosed to families and referred for genetic counseling and clinical validation. Variants of uncertain significance (VUS) in cancer predisposition genes represented 29% (n = 26) of germline findings and were identified in 21 (26%) participants. The majority of variants (62%, n = 56) were classified as benign or likely benign.

To describe the overall contribution of germline variants, we retrospectively assessed rare coding and splice site variants in 98 cancer susceptibility genes for all participants with germline sequencing (n = 81; Supplementary Table S2). Seventeen P/LP germline variants were identified in 16 individuals (20%; Fig. 2B). Excluding seven children with neurofibromatosis type I and pathogenic variants in *NF1*, all of which were known to their oncologist, nine individuals (12%) had P/LP germline variants in high- and moderate-penetrance genes. Six of ten (60%) P/LP variants were not known prior to study participation. Moderate-penetrance variants in *PALB2* and *APC* were each identified in one individual. Given other documented cancer history and/or indications for cancer risk management, recontact was sought for these families. Neither carrier of monoallelic variants in *MUTYH* and *NTHL1* were known to have a family history consistent with autosomal recessive *MUTYH*- and *NTHL1*-associated polyposis syndromes, and these findings were not disclosed to families.

Three of seven *NF1*-related tumors, including six plexiform neurofibromas and one peripheral nerve sheath tumor, showed secondary somatic genomic alterations in *NF1*. Pathogenic germline variants in *RB1* (n = 1), *SMARCB1* (n = 1), and *TP53* (n = 2) were similarly associated with secondary somatic events in a pontine glioma, rhabdoid tumor, and glioblastoma multiforme and osteosarcoma, respectively. Moderate-penetrance pathogenic variants in *CHEK2*, *PALB2*, and *APC* were not associated with loss-of-heterozygosity or a second somatic variant to implicate them in tumorigenesis. The *PALB2* germline variant was not associated with somatic signatures of homologous recombination deficiency (HRD) in a glioblastoma. No germline alterations were identified in any hematologic malignancy in this cohort.

### Identification of recurrent somatic mutations with therapeutic relevance

WGTA was conducted to identify somatic variants including single nucleotide variants (SNVs), structural variants (SVs), copy number variations (CNVs), and expression outliers. The most recurrent somatic genomic alterations in established tumor suppressor genes included mutations and/or copy losses in *TP53* (33% of cases), *ATRX* (24%), *RB1* (22%), *CDKN2A* (22%), and *CDKN2B* (20%), while mutations and/or copy

gains were found in oncogenes including *MDM4* (35%), *AURKA* (27%), *AKT3* (25%), *MET* (25%), *CDK6* (25%), *MYC* (24%), *EGFR* (23%), and *KRAS* (23%) (Fig. 3A). The most common SVs with both genomic and transcriptomic support identified in this cohort were those affecting *EWSR1* (*EWSR1-ATF1*, *EWSR1-FLI1*), *PAX3-FOXO1*, and *ETV6* translocations (*ETV6-NTRK3*, *ETV6-ZBTB44*). Therapeutically actionable somatic variants, as defined by the MTB, were identified recurrently in several genes encoding receptor tyrosine kinases (RTKs), cell cycle regulatory

genes, *PI3K/mTOR* and *RAS/MAPK* signaling pathways, and in epigenetic transcriptional regulatory genes.

In addition to genomic data, RNA sequencing identified genes with aberrant expression in pathways with therapeutic relevance. The expression profile of each sample was compared to datasets of normal tissue and disease comparator datasets, selected by tumor type and correlation of case-specific transcriptome data to all tumor types in the dataset. Outlier high expression of genes or signaling pathways

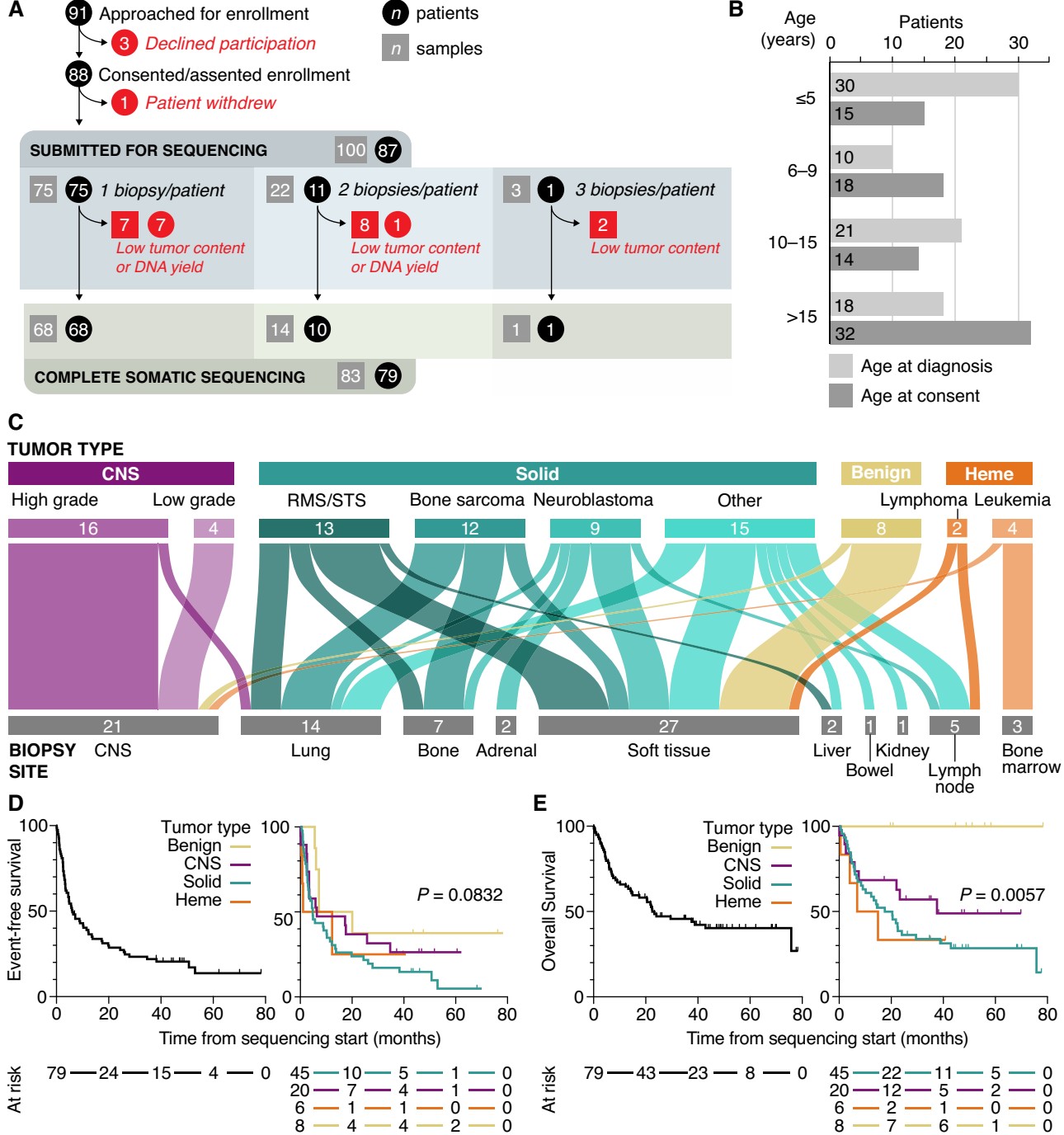

**Fig. 1 | Enrollment and tumor diagnosis details for participants in the pediatric POG study. A** Consort diagram depicting patient enrollment and sample progression including patient consent/assent, number of tumor biopsies conducted (includes archival tissue samples), and number of samples with completed WGTA (*n* = 83) for final study cohort (*n* = 79) participants; includes one participant with only WGS results. **B** Age of participants at diagnosis and enrollment. **C** Alluvial plot of patient tumor types for all successfully sequenced WGTA samples (*n* = 83) matched to primary biopsy site. **D, E** Kaplan–Meier plots of event-free and overall survival outcomes from sample sequencing start date for all pediatric POG study cohort participants (*n* = 79) and by tumor group. *P* values determined using two-sided log rank tests. CNS Central Nervous System, Heme hematological. Source data are provided as a source data file.

## Table 1 | Clinical characteristics and sequencing timelines for pediatric cohort

| Participant characteristics | N (#) | Range or proportion (%) |
|---|---|---|
| Age (years) | | |
| Median age at diagnosis | 8.8 | 0.3–20.7 years |
| Median age at enrollment | 13.8 | 0.5–21.2 years |
| Sex | | |
| Male | 46 | 58.2 |
| Female | 33 | 41.8 |
| Priority for biopsy per patient (n = 79 patients) | | |
| POG | 20 | 25.3 |
| Clinical | 56 | 70.1 |
| Both | 2 | 2.6 |
| Not applicable | 1 | 1.3 |
| Tumor type (n = 83 samples[a]) | | |
| RMS/STS | 13 | 15.7 |
| Bone sarcoma | 12 | 14.5 |
| Neuroblastoma | 9 | 10.8 |
| High grade CNS | 16 | 19.3 |
| Low grade CNS | 4 | 4.8 |
| Benign | 8 | 9.6 |
| Leukemia | 4 | 4.8 |
| Lymphoma | 2 | 2.4 |
| Other solid | 15 | 18.1 |
| Tumor category (n = 83 samples[a]) | | |
| Solid | 49 | 59.0 |
| CNS | 19 | 22.9 |
| Heme | 6 | 7.2 |
| Benign | 9 | 10.8 |
| Tumor sample sequencing time point (n = 83 samples[a]) | | |
| At initial diagnosis | 27 | 32.5 |
| At progression/relapse | 56 | 67.5 |
| Turnaround time to MTB meeting (days) (n = 83 samples) | | |
| 30–60 | 20 | 24.1 |
| 61–90 | 43 | 51.8 |
| 91–120 | 12 | 14.5 |
| 120+[b] | 2 | 2.4 |
| No MTB[c] | 6 | 7.2 |
| Median MTB turnaround time (days) | 71 | |
| Turnaround time to TGR results (days) (n = 83 samples) | | |
| 0–30 | 27 | 32.5 |
| 31–60 | 43 | 51.8 |
| 61–90 | 3 | 3.6 |
| 91–120 | 1 | 1.2 |
| No TGR | 9 | 10.8 |
| Median TGR turnaround time (days) | 34 | |

*RMS/STS* rhabdomyosarcoma/soft-tissue sarcoma, *CNS* central nervous system, *TGR* Targeted Gene Report, *MTB* Molecular Tumor Board.

[a]One patient had two tumors biopsied from different regions of the body; three patients had two samples sequenced at different times.

[b]One case was de-prioritized following the patient's death.

[c]Four patients died in <3 weeks from the time of sample receipt and a formal MTB was not scheduled.

encoding RTKs, cell cycle regulators, and components of *PI3K/AKT/mTOR* were most frequently observed (Fig. 3A). Among 79 participants, 73 (92%) had at least one expression-based alteration discussed at MTBs (Fig. 3B). In total, 116 expression outliers were considered actionable in 58 patients, of which 21% had moderate evidence (LOE3)

to support actionability, while 78% (90 outliers) were classified at lower levels of evidence (LOE4-5). Out of 173 therapeutically actionable findings, 33% were supported by DNA-level evidence alone, 38% were supported by RNA expression evidence alone, and 29% were supported by a combination (Fig. 3C, Source Data File 1). DNA alterations were integrated with gene expression data at the pathway level, providing support for the functional impact of genomic variants, and suggested potential targets for putative oncogenic drivers that were not directly targetable[19]. Among the therapeutically actionable findings, 48 (28%) were supported by alterations in multiple genes that interact or belong to the same pathway, such as copy loss of *CDKN2A* and high expression of *CDK6* in cell cycle regulation, or high expression of *IL-6* and *JAK1* in JAK-STAT signaling.

### Mutation signature analyses and genome stability evaluation
Beyond individual genes, the genome-wide signatures such as tumor mutation burden (TMB), mutation signatures, and genome stability were evaluated to characterize the tumor genome and support potential therapy targets.

Deconvolution of somatic mutational signatures defined in the Catalogue of Somatic Mutations in Cancer (COSMIC, version 3.2)[24] revealed that the global mutational landscape of pediatric tumors was largely characterized by clock-like mutational processes, reflecting exposure to single base substitution (SBS) signatures 1 and 5 (Fig. 4A). Few pediatric tumors showed contributions from signatures associated with exposure to UV light or tobacco, to which a subset of advanced and metastatic adult tumors showed strong exposure. APOBEC-mediated mutagenesis, a common endogenous mutational process in human cancers enriched in certain cancer types such as breast, lung, and liver cancer, did not show strong exposure in pediatric tumors[25]. While aberrant homologous recombination showed similar contributions to mutagenesis across pediatric and adult tumors, signatures of *MUTYH*-related base excision repair deficiency were enriched in pediatric soft-tissue sarcoma and neuroblastoma (Fig. 4B). A patient with *MYCN* amplified neuroblastoma with high mutation signature 3 (*BRCA*1/2) exposure (89th percentile in the pediatric cohort) had a germline heterozygous *PALB2* missense mutation (p.R663C), also heterozygous in tumor, supporting a PARP inhibitor therapy recommendation (LOE5B) which was not pursued. *PALB2* showed outlier high expression (99th percentile) as did other genes involved in HR including *BRCA1* (94th percentile), *BRCA2* (100th percentile), and *HERC2* (100th percentile) in this tumor.

The SV burden was calculated for each sample to evaluate genome stability. SV is also used to calculate HRD score[26] to detect deficiency in homologous recombination. The HRD scores in this cohort ranged from 0 to 45, which corresponds to the 0–92 (median 23) percentile when compared to the adult cohort of POG. High HRD scores were observed in osteosarcoma, in which chromosomal rearrangements are often observed[27]. PARP inhibitor therapy was supported (LOE5B), though not received, in a patient with a malignant peripheral nerve sheath tumor, high HRD score (98.9th percentile in pediatric cohort), high SV burden (244; 93rd percentile in the pediatric cohort), and a somatic homozygous *ATM* mutation (p.F2839L), along with germline *NF1* heterozygous loss of function mutation.

### Whole genome tumor mutation burden and immune signatures in pediatric cancers
Median whole genome tumor mutation burden (TMB) in the pediatric cohort was 1.56 (range 0.28–7.87) mutations/Mb, excluding formalin-fixed, paraffin-embedded samples and samples from patients with a prior allogeneic stem cell transplant. When compared to 570 adult advanced cancers studied in POG[21], excluding those with somatic mutations in the mismatch repair genes or post allogeneic bone marrow transplant, the median adult tumor TMB was 3.62 mutations/Mb (range 0.29–274), which was significantly higher than in children

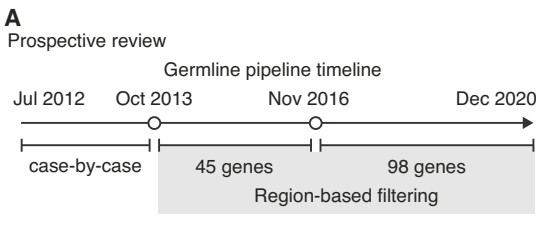

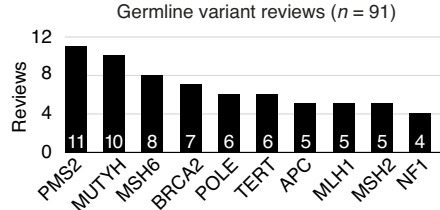

**Fig. 2 | Germline findings for participants in pediatric POG study. A** Evolution of the germline analysis pipeline in the pediatric POG cohort (left). Number of germline variants per gene reviewed prospectively by the POG Germline and Ethics Working Group during the study. **B** All pathogenic/likely pathogenic germline variants identified among 98 cancer predisposition genes within the pediatric POG cohort. SNV Single nucleotide variant, LOH loss of heterozygosity, POG Personalized Oncogenomics program. Source data are provided as a source data file.

($P$ = 5.67e−16). Coding TMB (Methods) was significantly correlated with TMB derived from the whole genome (R = 0.95, $p$ < 2.2 × 10−16, Spearman), and was lower than in our adult cohort (median 1.28 in the pediatric cohort, range: 0.08−7.44), though this did not reach statistical significance ($P$ = 0.13, Wilcoxon rank sum).

Transcriptome-wide assessment of tumor immune signatures, using CIBERSORT deconvolution of RNA data to derive immune cell scores, was undertaken for the cohort, excluding hematologic cancers or samples of lymph node origin (Supplementary Fig. 1). When compared to the adult POG pan-cancer cohort[21], pediatric tumors had lower CD8+ T-cell scores (Fig. 5A; $P$ = 0.0003, Wilcoxon rank sum) and total T-cell scores (Fig. 5B; $P$ = 0.0015) overall. High immune presence (>80th percentile in the cohort) was observed across a range of tumor types including tumors with *SMARCB1/A4* loss (chordoma, rhabdoid tumors), neuroblastoma, rhabdomyosarcoma and osteosarcoma, and high grade CNS tumors (Fig. 5C, D). No correlation between TMB and CIBERSORT CD8+ T-cell score (r = 0.11, $P$ = 0.34, Spearman) or total T-cell score (r = 0.003, $P$ = 0.97, Spearman) was observed. Similarly, no strong correlation

was observed between TMB and expression of several immune checkpoint genes, including *CTLA4* (r = 0.14, $P$ = 0.22, Spearman), *PD-L1* (r = −0.24, $P$ = 0.04), and *PD-1* (r = 0.15, $P$ = 0.21). The three cases with highest inferred immune presence (total T-cell scores) were all cases with *SWI/SNF* loss (rhabdoid tumor, chordoma) in which TMB was relatively low (Fig. 5B). Both chordomas were shown to have outlier high CD8+ T-cell scores in comparison with other pediatric cancer datasets including Treehouse, TARGET and a published cohort of rhabdoid tumors[28]. One chordoma patient subsequently achieved radiographic response to immune checkpoint inhibitor (ICI) therapy, and was explored further in a case study, confirming the high T-cell scores with IHC as well as high *PD-L1* expression[28].

## Treatment recommendations for therapeutically actionable findings

Among 79 participants, 173 therapeutically actionable findings were discussed at MTBs in 76 (96%) participants, of which 53 (70%) had multiple actionable variants identified. (Supplementary Data 1) Two

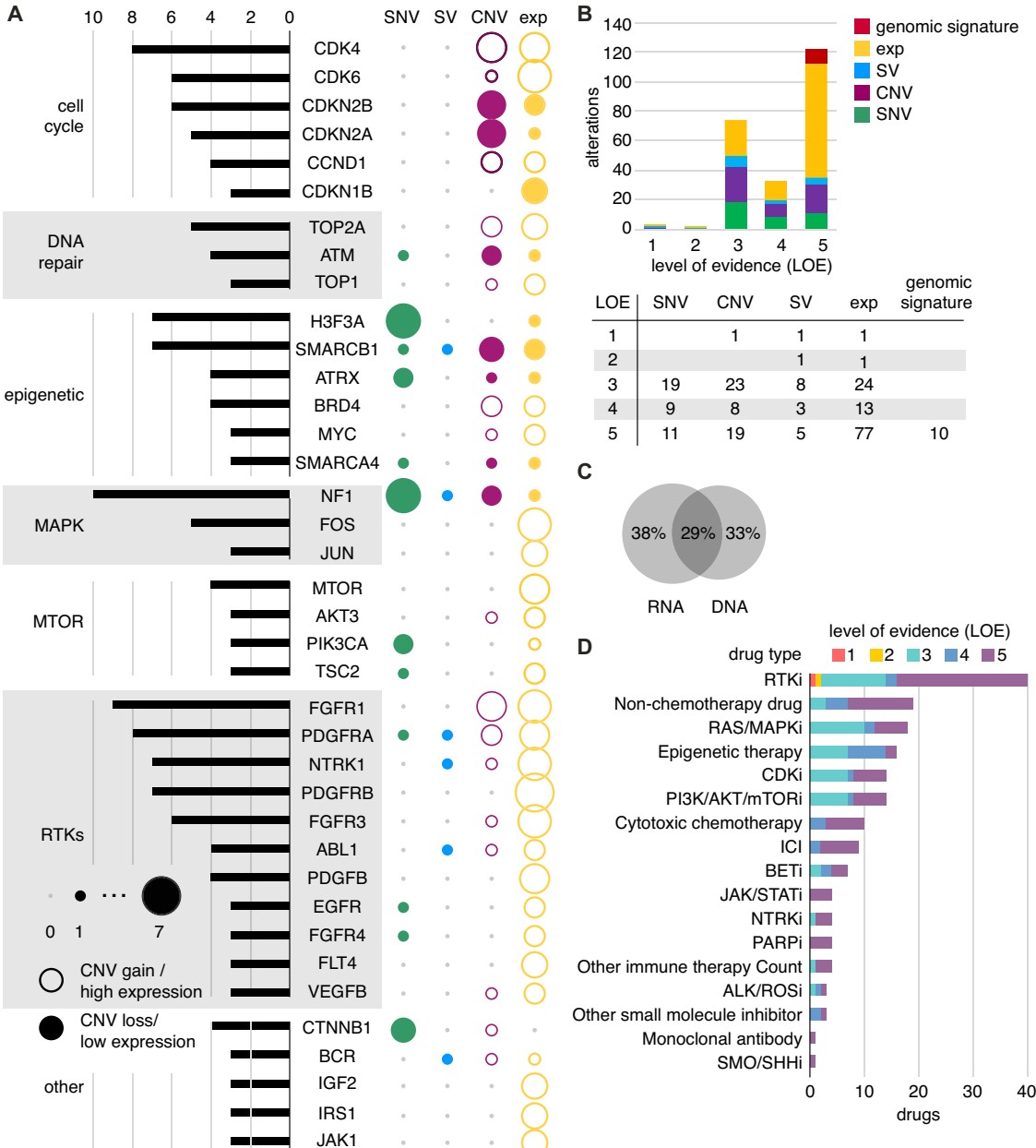

**Fig. 3 | Therapeutically actionable somatic findings for participants in pediatric POG study. A** Recurrent genes with therapeutically actionable alterations by variant type, grouped in pathways; each gene was reported in at least three patients; bar plot: gene list on the y-axis with number of patients on the x-axis; bubble plot: variant type of therapeutically actionable alterations (copy number variant, CNV; structural variant, SV; single nucleotide variant/insertions and deletions, SNV/ indels; expression outlier). The size of the bubbles is proportional to the number of patients. For gene expression, hollow bubbles indicate high outlier expression, and solid indicate low expression outliers. For CNVs, hollow bubbles are gains, and solid bubbles are losses. **B** Level of evidence (LOE, 1-5) for therapeutically actionable alterations, color coded by type of variants. **C** Therapeutically actionable alterations based on support from DNA and/or RNA sequencing data. **D** Categories of targeted drug therapy discussed at molecular tumor board based on somatic variants, color coded by level of evidence (LOE1-5). Source data are provided as a source data file.

participants had high-level therapeutically actionable variants (LOE1-2) that had not been identified clinically, and both were fusion events (*BCR:ABL* and *CCDC6:RET*; Fig. 3B). Fifty-three (31%) actionable findings discussed were supported by limited published clinical experiences (series or reports) or inclusion criteria for ongoing clinical trials (LOE3). The largest proportion of findings were only supported by preclinical evidence (14.8%, LOE4) or were putative biomarkers, including the majority of expression-based outliers (52.3%, LOE5; Fig. 3B).

Treatment recommendations supported by therapeutically actionable variants were discussed at MTBs. Drug categories discussed included RTK inhibitors (23%), RAS/MAPK pathway inhibitors (11%), non-chemotherapy drugs (11%), epigenetic therapy (9%), CDK inhibitors (8%), PI3K/AKT/mTOR pathway inhibitors (8%), cytotoxic chemotherapy (6%), and immune checkpoint inhibitors (Fig. 3D; 5%). In contrast to the high rate of therapeutic target identification from WTGA results, 29 out of 73 rapid TGR identified molecular variants, among which 16 variants were not known to the clinicians prior to the reports (22%) and 23 variants were classified as therapeutically actionable (32%). Pathognomonic driver oncogenic fusions that were not therapeutically actionable were also identified in the rapid TGR of four patients; all were previously known clinically.

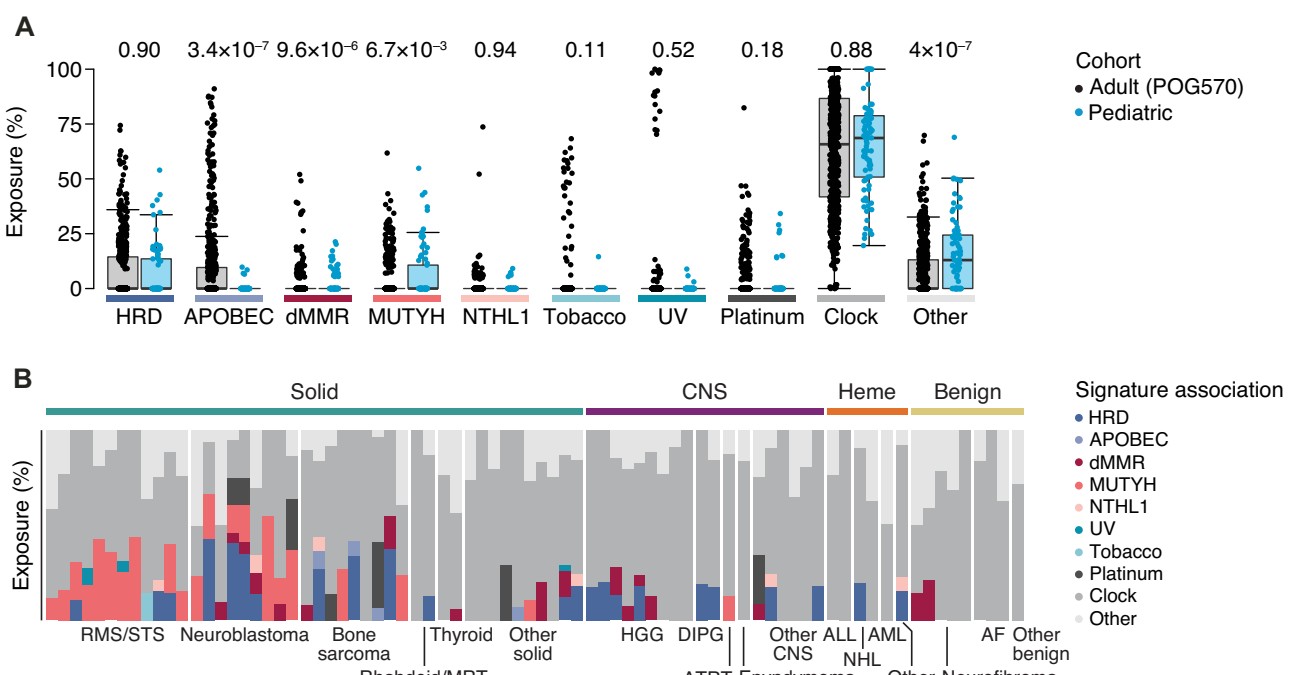

**Fig. 4 | Whole genome mutation signatures across pediatric tumors.**
**A** Proportion of small somatic mutations attributed to various mutational mechanisms in advanced or metastatic adult (*N* = 570) and pediatric (*N* = 81) cancers. Signature exposure was estimated by deconvolution of 96 single base substitution (SBS) signatures described in COSMIC v3.2 and grouped by proposed etiology: homologous recombination deficiency (HRD, SBS3 and −8), APOBEC-mediated mutagenesis (SBS2 and −13), mismatch repair deficiency (dMMR, SBS6, −14, −15, −20, −21, −26 and −44), MUTYH-related base excision repair deficiency (SBS18 and −36), NTHL1-related base excision repair deficiency (SBS30), tobacco exposure (SBS4 and −29), UV exposure (SBS7a, −7b, −7c, −7d and −38), platinum exposure (SBS17b, −31 and −35), and clock-like mutagenesis by spontaneous deamination of 5-methylcytosine to thymine (SBS1, −5 and −40). Boxplots display the median, and upper and lower quartiles with whiskers indicating minima and maxima. Statistics were derived using Wilcoxon rank sum tests. **B** Estimated contribution of SBS signatures by proposed etiology to pediatric tumors in POG. Tumor types listed in **B** are as follows: RMS/STS rhabdomyosarcoma/soft-tissue sarcoma, MRT malignant rhabdoid tumor, HGG high grade glioma, DIPG Diffuse intrinsic pontine glioma, ATRT atypical teratoid rhabdoid tumor, CNS central nervous system, ALL acute lymphocytic leukemia, NHL non-Hodgkin's lymphoma, AML acute myeloid leukemia, AF Ameloblastic fibroma. Source data are provided as a source data file.

## Outcomes of molecularly informed therapy trials

Eighty-three samples in 79 participants were reviewed at MTBs and 96% (*n* = 76 participants) had a potentially therapeutically actionable variant discussed, with 28 (35%) participants ultimately receiving molecularly informed therapy (Fig. 6A; Supplementary Data 2). Thirty-two molecularly supported therapies were pursued for 28 participants, with three participants receiving more than one therapy (Fig. 6B). Therapies were accessed by compassionate (*n* = 15), commercially available, off-label (*n* = 6), clinical trial enrollment (*n* = 6), and standard relapse therapy (*n* = 5) routes. Therapy was given as a single agent (*n* = 24) or as combination therapy (*n* = 8). Among the 28 therapy trials assessed for response, five achieved radiographic response (partial response (PR) *n* = 2, complete response (CR) *n* = 3) and ten had stable disease (SD) for ≥6 months (Fig. 5A), for an overall benefit rate of 54%. If patients with locally advanced, benign tumors were excluded (*n* = 3), the overall benefit rate was 48%. Among the 15 therapy trials with clinical benefit (CR, PR, or SD ≥ 6 months), eight were supported by moderate evidence (LOE3) at the time of MTB, while the remainder were supported by lower evidence (LOE4-5). Among participants achieving CR, all received combination therapy (mTOR inhibitor with chemotherapy in rhabdomyosarcoma with *PI3K* or *FGFR4* gain-of-function mutations; irinotecan combination therapy in neuroblastoma with an *ATRX* mutation). A participant with chordoma and outlier high immune signature T-cell scores, but low TMB, achieved PR to single agent ICI therapy (LOE5) and an additional patient with a *TRK* fusion achieved PR to TRK inhibitor therapy (LOE4B at the time of MTB). Among thirteen therapies which had no benefit, nine were for solid,

non-CNS tumors (LOE3B *n* = 2, LOE4B *n* = 2, LOE5B *n* = 5), three for CNS tumors (LOE5A *n* = 1, LOE5B *n* = 2), and one for acute myeloid leukemia (LOE3A). Among treated patients, the median EFS and OS from therapy initiation was 6.4 and 18.7 months, respectively, and this varied significantly by tumor group (Fig. 6C, D). Of the 32 therapy trials, the majority (*n* = 22) were for patients with solid, non-CNS malignant tumors. Among treated patients with solid tumors, the EFS from treatment start date did not differ by LOE (LOE3 versus LOE4-5) (Fig. 6E), but OS was superior for those patients receiving therapy with lower LOE (Fig. 6F; *P* = 0.039). Among 47 participants who received therapy recommendations but never accessed therapy, eight had no evidence of disease, eight were too unwell or deceased, and others declined therapy (*n* = 6). The remaining 22 participants either elected to undertake alternative therapies or drug access could not be obtained.

Among 32 therapies pursued, 11 were based on RNA evidence only and another 14 were supported by combined DNA and RNA alterations (Supplementary Table S4). Forty-four percent (*n* = 4/9) of patients assessed for response to therapies based on RNA only evidence achieved benefit (PR or prolonged SD), while 6 of 12 patients benefited from therapies based on combined DNA/RNA evidence. Two participants with high-level therapeutically actionable variants (LOE1-2) identified through POG (prior clinical fluorescence in situ hybridization test was negative for *BCR:ABL* in lymphoma; no prior test was done for *CCDC6:RET* in papillary thyroid carcinoma) did not receive targeted therapy due to patient choice and no evidence of disease.

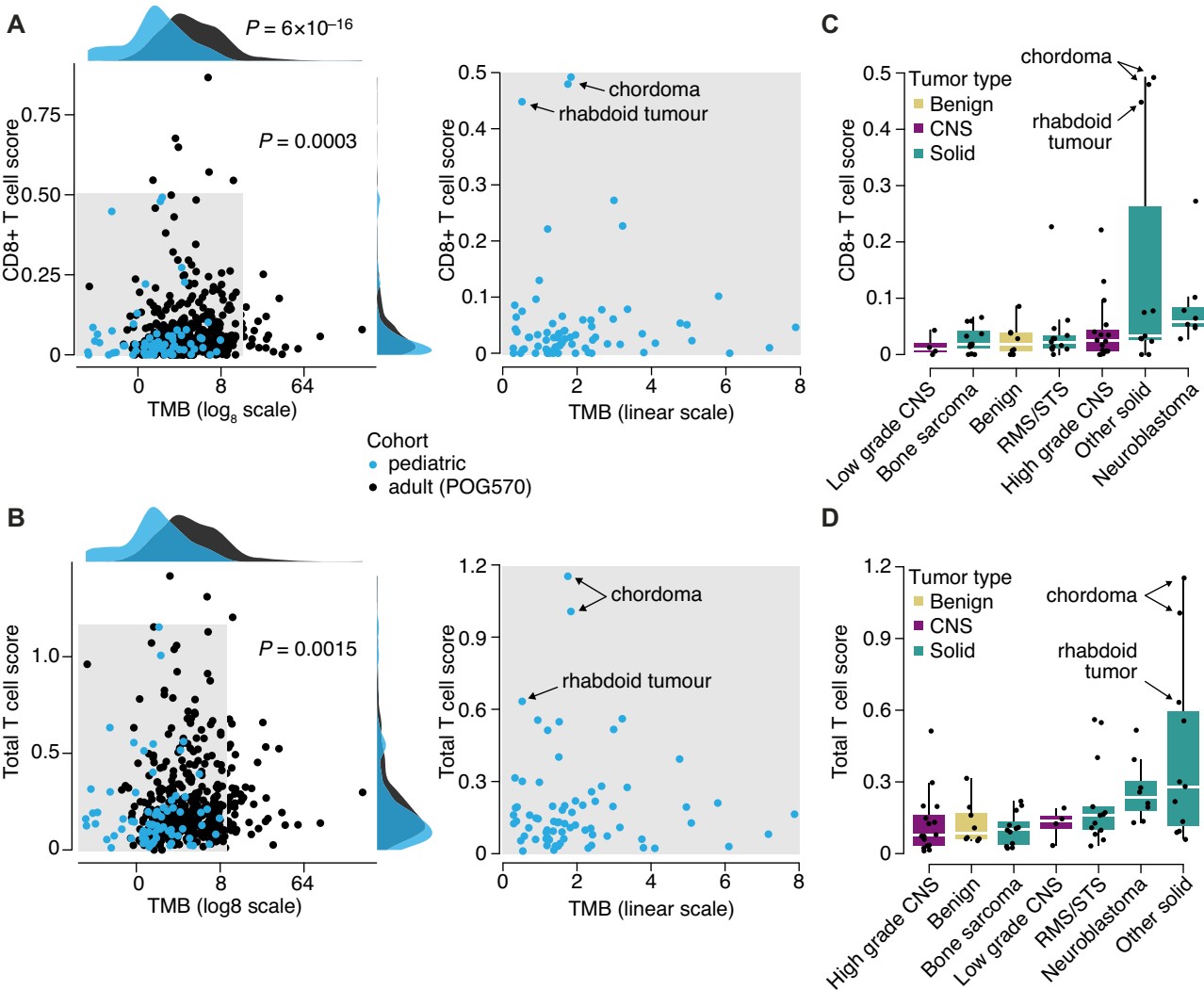

**Fig. 5 | Whole genome tumor mutation burden (TMB) and CIBERSORT immune signatures in pediatric POG patients and adult POG patients (POG570).** Pediatric tumors ($N = 72$) showed lower TMB and T-cell scores compared to adult tumors ($N = 423$). **A** CD8+ T-cell scores on the y-axis and TMB on the x-axis. **B** Total T-cell scores on the y-axis and TMB on the x-axis. Correlations shown are derived from Spearman tests, and comparisons between adult and pediatric used two-sided Wilcoxon rank sum tests. **C** CD8+ T-cell score by tumor sub-group. **D** Total T-cell scores by tumor sub-group. Boxplots display the median, and upper and lower quartiles with whiskers indicating minima and maxima. For panels **C** and **D**, sample sizes are as follows: High grade central nervous system (CNS), $n = 16$; Low grade CNS, $n = 4$; Benign, $n = 8$; Bone sarcoma, $n = 12$; Neuroblastoma, $n = 8$; Rhabdomyosarcoma/soft-tissue sarcoma (RMS/STS), $n = 13$; Other solid, $n = 11$. Source data are provided as a source data file.

## Discussion

Among our pediatric cancer cohort, 96% had putative or potentially therapeutically actionable somatic variants identified from integrated WGTA and discussed at an MTB, of which 62% of variants were identified at the DNA level (DNA alone or combined with RNA) and 38% were supported by transcriptome analyses only. Our virtual panel, the rapid TGR, identified actionable findings in only 32% of participants, of which most were known clinically prior to study enrollment. This supports conclusions drawn by other large pediatric cancer sequencing programs, such as MSK-IMPACT, that cancer NGS panels identify actionable alterations in the minority of pediatric advanced cancers and a more comprehensive approach holds great value and potential[16].

Excluding participants with locally advanced benign tumors, the clinical benefit rate for those receiving molecularly informed therapy in our cohort was 48% and the objective radiographic response rate (CR, PR) was 20%. In a comparable pediatric cancer cohort in Australia, 32% of patients received molecularly supported therapy and 31.4% achieved radiographic response, with another 40% achieving SD, which was defined as stability at a minimum interval of 6 weeks from

initiation of therapy[18]. The Australian cohort may have benefited from more frequent utilization of combination therapies in 42% of their treated patients. They also reported higher tier actionable variants more frequently, though this is likely related to differences in the level of evidence schema and highlights the need for standardization of criteria. The INFORM pediatric cancer cohort utilized a seven-tier level of evidence schema and 85.9% of patients had at least one actionable target, among which 33% received targeted therapy, with only 3.8% enrolled in a targeted therapy clinical trial[17]. Though they do not describe radiographic response or clinical benefit rates, there was no difference in survival outcomes comparing treated versus untreated patients overall.

A strength of our genomics pipeline is the routine integration of whole transcriptome sequencing to identify expression outliers, tumor genomic signatures, and evaluation of the immune microenvironment for therapeutic decision-making. Signals of outlier high gene expression in key cancer-related intracellular signaling pathways were frequently identified in our pediatric cohort and expression data often supported DNA-based observations providing additional evidence of

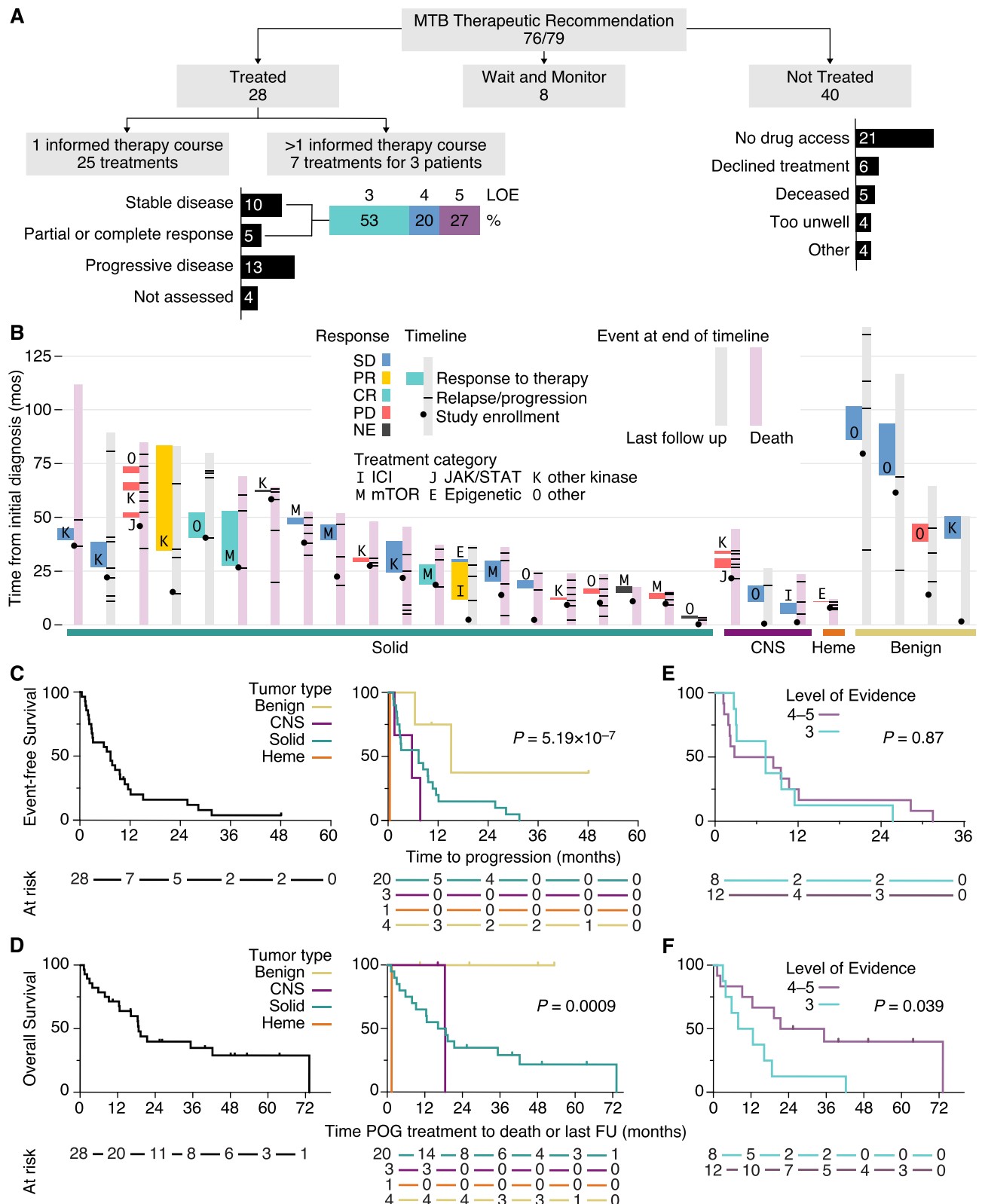

**Fig. 6 | Therapeutic recommendations and outcomes for participants in the pediatric Personalized Oncogenomics (POG) study. A** Clinical action flowchart depicting the number of participants from the final study cohort (n = 79) who received therapeutic recommendations by the molecular tumor board (MTB) (n = 76) and were either treated (n = 28), untreated (n = 40), or under surveillance (n = 8). **B** Among patients receiving therapy, Swimmer plot depicting full disease trajectory from diagnosis including disease type, response status (CR complete response, PR partial response, SD stable disease, PD progressive disease, NE Not

evaluable), and key milestone events (consent date, relapse or progression, date of last follow-up or death). **C** Event free survival (EFS) from initiation of POG-directed therapy for treated patients and grouped by tumor type. **D** Overall survival from initiation of POG-directed therapy among treated patients and grouped by tumor type. **E** EFS grouped by treatment level of evidence (LOE). **F** OS grouped by treatment LOE. Statistics for outcome were derived using two-sided log rank tests. Source data are provided as a source data file.

pathway activation downstream of a putative oncogenic driver. Comparative transcriptome-based analyses within a precision medicine framework are likely informative in a greater proportion of cases, than if analysis is limited to DNA-based WGS alone[19,29]. In our affiliated adult pan-cancer cohort, integrated analyses resulted in high actionability rates with RNA expression data contributing to most WGTA-informed therapies (67%)[22].

Deconvolution of bulk tumor transcriptome data for the prediction of immune cell presence provides insight into the local immune environment, which may have therapeutic implications. ICI therapy has an established role in a number of adult malignancies with underlying mismatch repair deficiency or high TMB[30]. Pediatric cancers have a lower burden of somatic mutations and trials of ICI therapies in unselected pediatric refractory solid tumors have shown limited efficacy[31,32]. A rare subset of pediatric solid tumors including malignant rhabdoid tumors, chordomas and epithelioid sarcoma which typically have low TMB, outlier high immune infiltration and mutations affecting the SWI/SNF complex (*SMARCB1/A4*), have been among rare responders to ICI[33]. In our cohort, patients with these tumor types demonstrated outlier immune scores by CIBERSORT and low TMB. This, along with overexpression of *brachyury*, informed a compassionate access therapy trial of an ICI for a child with a poorly differentiated chordoma, who achieved radiographic response after three cycles[28]. We also observe CD8+ and total T-cell immune infiltration outliers among other tumor types including neuroblastoma, soft tissue sarcoma and high-grade CNS tumors (Fig. 5C, D). Immuno-transcriptomic analyses in large cohorts of pediatric solid tumors including neuroblastoma and sarcoma have demonstrated subsets with prominent CD8+ T-cell infiltration, correlation of immunomodulatory gene expression with CD8+ T-cell scores, and potential impact on survival outcomes[34].

Mutation signatures and HRD scores were also considered as supportive evidence for therapeutic actionability in multiple cases. A patient with *MYCN* amplified neuroblastoma who had *BRCA1/2* mutation signature exposure had a germline SNV in *PALB2*, and a somatic missense SNV in *HERC2*. This *PALB2* mutation was classified as a VUS in ClinVar but has been reported in patients with breast and colorectal cancer, and *PALB2* mutations confer susceptibility to neuroblastoma[35–37]. This patient's tumor also harbored a *HERC2* heterozygous mutation in the functional domain (p.G3027R). *HERC2* is an E3 ubiquitin ligase that regulates ubiquitin-dependent degradation of DNA repair proteins including *BRCA1*, and was an expression outlier in our patient's tumor, further supporting therapeutic targeting of this pathway which ultimately was not pursued[38].

Among 32 molecularly supported therapies, only eight (25%) involved combination regimens. The three patients who achieved CR all received combination therapies which are considered standard salvage therapy approaches for their disease type[39,40]. In these cases, the value of tumor genomics may be in utilizing molecular selection to help identify potential responders to established regimens. Both patients with rhabdomyosarcoma and actionable somatic gain-of-function mutations predicting response to mTOR inhibition achieved CR to temsirolimus with vinorelbine and cyclophosphamide, compared to 5 of 42 participants who achieved CR in the molecularly unselected pediatric phase II study arm[39].

Although 35% of our participants received molecularly informed therapies, only six patients enrolled on a clinical trial, similar to other pediatric cohorts[17,18]. Of the six patients, two required travel outside of Canada (Germany, U.S.) and a third was only able to access an innovative therapy following approval of a Health Canada single patient protocol of a combination regimen[41]. Outside the context of clinical trials, therapies were only feasible if pediatric dosing and toxicity data were available, and therapies in appropriate formulation were accessible via compassionate or off-label access routes. Despite legislation mandating pediatric drug development plans for innovative agents in

certain jurisdictions[42,43], the majority of young people receiving molecularly-targeted cancer therapies in Canada and around the world are doing so outside the context of a traditional clinical trial. There is a pressing need to capture systematic outcome data for all innovative therapies and efforts to address this prospectively are ongoing internationally (NCT04477681).

When integrating molecular results into patient care, it is important to consider that levels of evidence are constantly evolving, and therapy recommendations need systematic reassessment over time. An early patient in our cohort had an *NTRK2* fusion which was classified as LOE4B for pan-TRK inhibitor therapy at the time of sequencing. The patient subsequently accessed larotrectinib via a pediatric clinical trial (LOE3A)[44] and *NTRK* fusions were only later approved to predict response to larotrectinib (LOE1A)[45]. The need for continuous integration of new knowledge and re-assessment of evidentiary support for therapeutic actionability has also been highlighted by the pediatric INFORM registry, in which 21.6% of patients had a change in priority level of actionable targets over the course of the trial[17]. Among our patients with solid, non-CNS tumors who received molecularly informed therapy, those directed by lower levels of evidence (LOE4-5 versus LOE3) had similar EFS (Fig. 6E) and superior OS (Fig. 6F). This could be due to LOE reclassification over time or the possibility of selection bias; patients with a more indolent disease may have been more likely to obtain "innovative" therapies.

The increasing use of germline sequencing in pediatric oncology has revealed that P/LP variants in cancer predisposition genes occur in 7.5–18% of children unselected for family cancer history[14,15,17,18,46]. Our cohort had a similar prevalence, and an estimated 50–70% of these continue to be missed clinically in our cohort and others despite availability of clinical tools to identify high-risk patients[17,47,48]. Our data support universal availability of testing for germline cancer predisposition variants in pediatric oncology due to high prevalence and demonstrated benefits to the patient, family, and health-care system following initiation of cancer surveillance programs.

Limitations of our study include cohort heterogeneity and sample size. In order to study recurrent genomic alterations and responses to targeted treatments at a population level, larger sample sizes and data sharing are necessary. Hematologic malignancies were under-represented, as seen in other pediatric cancer cohorts[17,18], due to difficulties acquiring suitable samples with adequate tumor content, rapidity of disease progression, and availability of alternative therapeutic strategies. These issues may be addressed with advances in single-cell sequencing and streamlined approaches to genomics and bioinformatics. Turnaround time and timing of WGTA in the disease trajectory are important considerations for patients with aggressive disease in obtaining maximal benefit. Despite having explicit eligibility criteria of anticipated life expectancy ≥3 months at enrollment, 24% of our patient cohort died prior to or within 3 months of their MTB, limiting utility. A shift to WGTA earlier in disease trajectories will be important to ensure that we optimize the incorporation of actionable findings into patient care. A move towards an increasingly streamlined WGTA workflow will further optimize result utility and is evident in recent pediatric precision oncology cohorts with reported TATs of <2 weeks[16].

Comprehensive WGTA is feasible for pediatric and adolescent relapsed, refractory, or poor prognosis cancers in clinically meaningful timeframes. Integration of transcriptome data provides support for actionability of DNA-based targets and can provide insight into molecular pathway activation, increasing the likelihood of therapeutically actionable results. Access to molecularly informed therapies remains a major hurdle that demands a multi-faceted solution including expanded access to collaborative clinical trials and facilitation of drug access pathways with a mandate to prospectively capture and share outcome data internationally. Although panel sequencing has an established role in pediatric oncology diagnosis, risk

stratification, and therapeutic decision-making, there is evidence to support additional clinical utility of a broader WGTA approach. Methods to routinely incorporate RNA-based analyses for a multi-omics approach to poor prognosis cancer will likely prove informative for both diagnostic and therapeutic actionability, though challenges remain regarding routine clinical validation and the need for further evidence generation.

## Methods

### Ethical oversight, inclusion criteria, and consent

This research complies with all ethical regulations and the study protocol was approved by the University of British Columbia, Children's and Women's Research Ethics Board (#H13-01640). All participants gave written informed consent/assent, according to CARE guidelines and in compliance with the Declaration of Helsinki principles. For participation in the PedsPOG program, eligible patients were identified by their treating oncologist. Inclusion criteria required patients to have a histologic diagnosis of cancer which was relapsed, refractory, or poor prognosis (predicted overall survival (OS) < 30%). Participants were followed by a tertiary care pediatric oncology program, had predicted survival >3 months, and Lansky or Karnofsky performance status >50. Participants were excluded if they did not have or were not willing/able to obtain a tumor sample. Informed consent and assent were obtained for clinical data and tumor tissue submission, study biopsy, peripheral blood collection, and somatic and germline WGTA. Fresh frozen tumor samples, at time of enrollment, were submitted following either a clinically indicated surgical procedure or a minimally invasive study biopsy. A blood sample (or a skin biopsy for patients with prior allogeneic bone marrow transplant) was submitted as a source of germline DNA. A comprehensive consent from legally authorized representatives and age appropriate assent were undertaken by a study pediatric oncologist and past family medical history was reviewed in detail. The possibility and implications of identification of germline cancer predisposition variants, along with incidental findings, were discussed. Optional consent for re-contact in case of future re-classification of variants of uncertain significance, future research, and for broad data-sharing was also obtained. Re-consent procedures were implemented for adult participants at the age of majority (19 y). Sex was recorded as coded in health records and reported in aggregate as a descriptive characteristic.

Clinical, pathological, and genomic results, along with an overview pathway diagram, were discussed at a multi-disciplinary MTB meeting with the study team, including pediatric and medical oncologists, bioinformaticians, and scientists from Canada's Michael Smith Genome Sciences Centre. The meetings reviewed actionable variants, annotated by LOE, with reference to a shared, prospectively updated Knowledge Base, medical literature, and clinical trial availability (www.clinicaltrials.gov). The MTB results, including P/LP germline variants, were relayed to participants and families by their treating oncologist. Genetic counseling referrals and clinical validation testing were coordinated for participants with germline findings. All subsequent therapy decisions following result disclosure were at the discretion of the clinical team, patient, and family. A subset of participants elected to pursue molecularly informed therapy when a clinical trial or age-appropriate recommended phase 2 dose data were available and access to therapy could be obtained.

### Whole genome and transcriptome sequencing

Tumor genomes were sequenced on Illumina HiSeq 2500 using v3 or v4 chemistry and paired-end 125 base reads, or on HiSeqX using v2.5 chemistry and paired-end 150 base reads. Transcriptomes were sequenced on Illumina HiSeq2500, or on NextSeq500 using v2 chemistry. Mean coverage depth was 45X for the blood samples, and 91X for fresh frozen tumor samples.

### Detection of somatic alterations

Sequence reads from normal and tumor whole genome libraries were aligned to the human reference genome (hg19) using the Burrows-Wheeler Alignment tool (v0.5.7 for up to 125 bp reads and v0.7.6a for 150 bp reads)[49]. Regions of somatic CNV and losses of heterozygosity were identified using the Hidden Markov model-based approaches CNAseq (v0.0.6)[50] and APOLLOH (v0.1.1)[51], respectively. Tumor purity and ploidy were identified using in-house scripts followed by manual review. Somatic SNVs were identified using two approaches: (1) putative somatic variant calls from SAMtools (v0.1.17)[52] with subsequent scoring by machine-learning based MutationSeq (v1.0.2 and v4.3.5)[53], and (2) identification and scoring with the joint caller Strelka (v1.0.6)[54]. Small indels were identified using Strelka with QSI ≥ 15. Total genomic TMB was the total number of SNVs and indels per sample and both were calculated for the whole genome. Variants were annotated to genes using SNPEff (v3.2) with the Ensembl database (v.69)[55]. SVs in DNA and RNA sequence data were identified using the assembly-based tools ABySS (v1.3.4) and TransABySS (v1.4.10)[56,57]. Putative SV calls identified from the DNA and RNA sequences were annotated against constitutional DNA to provide somatic and germline structural variant calls. The number of SVs identified from DNA was used as structural variant burden to evaluate genome stability.

### Mutation signature analysis and HRD score calculation

Somatic mutational signatures defined in COSMIC (v2 and v3.2)[24] were evaluated for somatic SNVs and indels identified by Strelka using SigProfilerAssignment (v0.0.31)[58]. Homologous Recombination Deficiency (HRD) scores were computed using the R package HRDtools[26] as the arithmetic LOH, telomeric-allelic imbalance, and large-scale state transitions scores, determined on the basis of published guidelines[59].

### Gene expression analysis

RNA sequencing (RNA-Seq) reads were analyzed with JAGuaR (v.2.0.3)[60] to include alignments to a database of exon junction sequences and subsequent repositioning onto the genomic reference hg19. RNA expression was quantified using in-house scripts as reads per kilobase per million mapped reads (RPKM). Gene annotations were based on the Ensembl database (v69)[55]. Gene expression was evaluated by comparing to publicly available transcriptome sequencing data from normal and tumor tissues, including data from Illumina BodyMap 2.0 (https://www.ensembl.info/2011/05/24/human-bodymap-2-0-data-from-illumina/), the Genotype-Tissue Expression (GTEx) Project (https://gtexportal.org/home/), The Cancer Genome Atlas (TCGA)[61], Treehouse Childhood Cancer Initiative[62], and the TARGET program[63]. For each case, one or multiple datasets were selected as disease comparators (from tumor tissues such as in TCGA and TARGET) and normal comparators (from normal tissues such as in Illumina BodyMAP and GTEx) based on diagnosis and Spearman correlation. Percentiles are calculated for each gene among comparators: ≥90th percentile was considered as high expression, ≤5th percentile was considered as low expression. A within-sample expression rank was also calculated to infer significance of outlier gene expression levels. For cases without matched comparators, an internal rank of top 5% was considered as high expression.

Expression data from non-hematologic malignancies (and excluding samples biopsied from lymph nodes) were analyzed using CIBERSORT R Package (v1.04) for immune cell deconvolution[64]. The LM22 cell subtype signature, composed of 547 genes, was used to predict the presence of 22 immune cell subtypes in each RNA-Seq sample. CIBERSORT was run without quantile normalization with 1000 permutations on RPKM data across all samples to generate absolute scores for each cell type. Total T-cell score was computed as the sum of all CIBERSORT T-cell scores excluding regulatory T-cells. CIBERSORT

data for this pediatric cohort was compared to an adult advanced pan-cancer cohort sequenced on the same pipeline[21].

## Germline variant identification

Germline variant calling was performed as described previously[21]. Briefly, SNVs and indels were called using SAMtools (v0.1.17)[65], and copy number and structural variants were called using Control-FREEC (v5)[66] and DELLY (v0.7.3)[67], Manta (v1.0.0)[68], ABySS (v1.3.4)[56], and MAVIS (v2.1.1)[69], respectively. SNVs and indels were annotated using SNPEff (v4.1)[70], and region-based filtering was performed for lists of 45 genes associated with high- and moderate-penetrance germline cancer susceptibility between 2013 and October 2016 and 98 genes from November 2016 to December 2020 (Supplementary Table S2). Custom scripts were used to prioritize variants for review by a clinical molecular geneticist. Variants were flagged for review by a POG Ethics and Germline working group which included clinical geneticists, bioinformaticians, and clinical team members, prior to disclosure to treating oncologists[71]. P/LP variants were disclosed by treating oncologists and patients/families were offered referrals to the most appropriate medical genetic clinic for genetic counseling and to discuss options for clinical validation and family testing, where applicable. Clinical confirmation was performed by site-specific testing according to standard clinical protocols at a CLIA/CAP-accredited laboratory, depending in part on eligibility for provincially-funded testing and test availability.

Small germline variants within the 98 genes were prioritized for retrospective review across all cases by region-based filtering and annotated using ANNOVAR (version 2018-04-16)[72] and InterVar (version 2.0.2)[73]. Variants with P/LP assertions in ClinVar (version 2019-03-05)[74], P/LP predictions by InterVar, and non-synonymous variants with an allele frequency ≤0.5% in the Genome Aggregation Database (gnomAD) (https://gnomad.broadinstitute.org/)[75], were prioritized for review. Germline CNVs and SVs predicted to overlap with coding regions in any of the 98 genes were also prioritized for review. All candidate variants were reviewed in Integrative Genomics Viewer[76]. P/LP germline variants that were not identified at the time of case analysis were reviewed with the POG Ethics and Germline working group and disclosed to the treating oncologist where the finding was determined to be of potential clinical significance. Monoallelic variants in recessive genes were not reported unless there was a concerning family history indicating a risk for recessive disease.

## Integrated analysis of whole genome and transcriptome sequencing data

A rapid TGR was created and disseminated based on exact matches to known oncogenic mutations and gene fusions (Supplementary Table S3), collected in a Knowledge Base (GraphKB)[77], and returned to clinicians. Subsequently, genomic data (SNV, CNV, SV) and transcriptomic data (SV, expression) are integrated at both gene level and pathway level. Genes that directly or indirectly interact in the same pathway are analyzed as a unit and visualized in a pathway plot (Supplementary Fig. 2). Potential cause-effect links are established between alterations in upstream regulators and expression level of downstream effectors, which are integrated to identify pathway level aberrations and therapeutic targets. Results of integrated WGTA were compiled into a report using an open-source reporting platform[77], incorporating expert curation encompassing literature review, creation of pathway visualizations, and a summary of potentially targetable alterations. The GraphKB and Integrated Pipeline Reports (IPR) reporting platform are described and available for download at https://github.com/bcgsc/pori. WGTA reports were discussed in an MTB meeting, and variants offering a potentially feasible therapy option were defined as therapeutically actionable variants. Actionable variants were categorized using a Canadian pediatric precision oncology classification framework

(Supplementary Table S4) and LOE was assigned based on evidence available at the time of MTB.

## Clinical data and survival analyses

Baseline demographics, cancer type, prior therapy, known molecular aberrations, and response and survival outcomes were abstracted from clinical charts with a follow-up censor date of November 1, 2020. PFS and OS for all participants were calculated from date of sample receipt for sequencing to the date of event or last follow-up. An event was defined as the date of disease progression, relapse, or death from any cause. Molecularly informed therapy was defined as any therapy that was listed as a potential treatment recommendation on the genomic report and discussed at the MTB. Although outcome data was collected, molecularly informed therapies were not part of the POG study and all treatment decisions were made in the context of routine clinical care. Among participants who received molecularly informed therapies, PFS and OS were calculated from the start date of the therapy to the date of event or last follow-up. PFS and OS were estimated using the Kaplan-Meier method and compared by log-rank tests at $P = 0.05$ significance level. Response was assessed retrospectively by review of radiographic disease response assessments with application of Response Evaluation Criteria in Solid Tumors (RECIST)[78], Response Assessment in Neuro-Oncology Working Group (RANO)[79], or International Neuroblastoma Response Criteria (INRC)[80] criteria, as applicable. Clinical benefit was defined as complete response (CR), partial response (PR), or stable disease (SD) for ≥6 months. Patients who did not undergo disease evaluations were not evaluable for response.

## Statistics and reproducibility

No statistical method was used to predetermine sample size. The experiments were not randomized and the investigators were not blinded to allocation during experiments and outcome assessment. Any sample exclusions for analyses, including justification for exclusion, are reported in the relevant section of the Methods or Supplementary Note. Unless otherwise stated, all statistical tests were performed in R and P values stated reflect two-sided tests.

## Reporting summary

Further information on research design is available in the Nature Portfolio Reporting Summary linked to this article.

## Data availability

The tumor WGS and RNA-Seq raw data generated in this study have been deposited in the European Genome-phenome Archive [https://ega-archive.org/studies/EGAS00001006967]. Three other patients have been previously deposited and can be accessed at https://ega-archive.org/datasets/EGAD00001008012, https://ega-archive.org/datasets/EGAD00001008013 and https://ega-archive.org/datasets/EGAD00001004712. The WGS and RNA-Seq data are available under controlled access to ensure strict confidentiality. Access can be obtained by submitting a request to our Data Access Committee [https://ega-archive.org/dacs/EGAC00000000011]. Publicly available transcriptome sequencing data from normal and tumor tissues that are used for gene expression analysis is available at, Illumina BodyMap 2.0 (https://www.ensembl.info/2011/05/24/human-bodymap-2-0-data-from-illumina/), the Genotype-Tissue Expression (GTEx) Project (https://gtexportal.org/home/), The Cancer Genome Atlas (TCGA,https://portal.gdc.cancer.gov/), Treehouse Childhood Cancer Initiative (https://treehousegenomics.soe.ucsc.edu/public-data/), and the TARGET program (https://www.cancer.gov/ccg/research/genome-sequencing/target). Pediatric sequencing data was compared to our adult pan-cancer cohort on POG, for which data has also been deposited in the European Genome-phenome Archive

(Accession # EGAS00001001159). Source data are provided with this paper.

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

## Acknowledgements

This work would not be possible without the participation of our patients and families, the POG team, the Genome Science Centre platform, and the generous support of the BC Cancer Foundation, BC Children's Hospital Foundation and Genome British Columbia (project B20POG; M.A.M., J.L.). The results published here are in part based upon data generated by the following projects and obtained from dbGaP (http://www.ncbi.nlm.nih.gov/gap): The Cancer Genome Atlas managed by the NCI and NHGRI (http://cancergenome.nih.gov); Genotype-Tissue Expression (GTEx) Project, supported by the Common Fund of the Office of the Director of the National Institutes of Health (https://commonfund.nih.gov/GTEx). We also would like to thank the Treehouse Childhood Cancer Initiative[48] for sharing their pediatric cancer transcriptomic data, the TARGET program[49] for sharing their expression data, and the PRO-FYLE project[11] for their collaborative effort in optimizing our core genomics/bioinformatics pipeline and for co-development of our level of evidence schema. We also acknowledge contributions towards equipment and infrastructure from Genome Canada and Genome BC (projects 202SEQ, 212SEQ, 262SEQ; M.A.M., S.J.M.J. 12002; M.A.M., S.J.M.J.,

J.L.), Canada Foundation for Innovation (projects 20070; M.A.M., S.J.M.J., 30981; M.A.M., S.J.M.J., J.L., 30198; M.A.M., 33408; M.A.M., S.J.M.J., 42362; M.A.M., S.J.M.J. and 40104; S.J.M.J.) including the CGEN platform (35444; S.J.M.J.) and the BC Knowledge Development Fund. We are immensely grateful to Patrick Sullivan and the Team Finn Foundation for funding our initial cohort of five patients.

## Author contributions

R.J.D., S.R.R., M.A.M., J.L., S.J.M.J., Y.S., and P.C.R. conceptualized the study. Formal analyses were performed by R.J.D., Y.S., S.R.R., E.T., K.D., L.M.W., E.P., M.B., C.C., E.C., M.J., Y.M., R.A.M., A.J.M., K.W. and Y.Z. Data were collected by K.L.M., E.C., R.A.M., A.J.M., Y.Z., Y.M., C.C., M.B., K.W. and M.J. Provision of patient samples and curation of patient data was conducted by R.J.D., Y.S., S.R.R., C.D., A.F.L, E.P., J.M.T.N., S.A., and M.K. The original draft was written by R.J.D., Y.S., S.R.R., K.D., S.A., M.A.M., J.L., S.J.M.J., E.T., L.M.W., E.P., and J.M.T.N. reviewed and edited the manuscript. Data visualization was conducted by R.J.D., Y.S., E.T., K.D., L.M.W., J.M.T.N., S.A., and M.K. Project management and co-ordination was performed by R.J.D., S.R.R., M.A.M., J.L., S.J.M.J., A.F.L., C.D., L.A., A.V., S.S.Y., K.A.S., and A.F.

## Competing interests

The authors declare no competing interests.
