## [Peer Review File · Nature Communications]

REVIEWER COMMENTS

Reviewer #1 (Remarks to the Author): Expert in paediatric cancer clinical research and genomics

The authors report on results from the Pediatric Personalized OncoGenomic cohort, the adult counterpart was reported in 2020. Following consent, tumor and matched normal tissue underwent whole genome and transcriptome profiling. Pediatric cancer genomic profiling by next generation sequencing has been previously published as referenced by the authors. However, there are some elements in this program that have not been a focus of prior pediatric efforts. Specifically, gene expression profiles and immune profiles. Overall the study is well done with interesting results.

Major Comments

1. Cohort is relatively small with a paucity of hematologic malignancies despite the high frequency of these malignancies in pediatrics which is a major weakness of the manuscript.
2. Given prior pediatric studies did not include gene expression in their studies but rather focused on somatic and germline SNV/Indel/CNA/SV, I would recommend expanding this component of the manuscript. The clinical outcomes description is described based on LOE, however gene expression is included in LOE3-5. What was the contribution of expression signatures to patients receiving targeted therapy and the response? In many cases, the somatic mutations/fusions dictate the gene expression. Were there cases where expression signatures were found in the absence of a pathogenic mutation in a given pathway leading to use of a targeted agent and clinical response? This would argue that expression signatures are more than just a complement to the detection of mutations.
3. Supplementary Table targeted therapies would benefit from adding the molecular mutation/gene expression signature that led to the recommended targeted drug as well as the response. Currently the table is limited to the level of evidence and there is no response information.

Minor Comments

1. In the last section of results, 53.3% of the 28 patients that pursued molecularly targeted therapies had a PR/CR/SD. The discussion specifies 48% of patients benefitted. Please clarify this discrepancy.

Reviewer #2 (Remarks to the Author): Expert in paediatric cancer clinical research, genomics, and tumour immune microenvironment

The manuscript from Deyell and colleagues describes the whole genome and transcriptome analysis of paediatric patients enrolled in the POG study between 2013 and 2019 in British Columbia in Canada. In this manuscript the authors report on the 79 participants who received results from this study. The primary strength of the work is that it is one of only a few reports of genomics analysis being applied across all cancer subtypes (so-called "pan-cancer") and of genomics being applied in a clinical setting, that is with a primary purpose of informing clinical decision making. In this context, this manuscript is an important contribution to the bodies of work assessing the clinical impacts of genomics in paediatric cancer.

The weaknesses of the manuscript are the relatively small cohort size (which is not changeable and which is a relatively minor issue for this reviewer) and the very descriptive style of manuscript (which is changeable and more important). Whilst it is not this reviewer's intention to impose any sort of writing style on the authors, one suspects that plain descriptions of the cohort, and range of mutations and other features are likely to be less interesting to readers (because things like this have already been published) than elements of the work which expand on the potential utility of genomic analysis of cancers (for example the immunoprofiling), ways in which different data types (WGS and RNA) are operationally integrated in developing patient reports and measures of the potential impact of this program. For example, In Figure 1D and E the authors show the Kaplan-Meier curves for the cohort dependent on the diagnostic category and in Figure 5 C-F they show the curves for the POG-therapy treated patients. One imagines that the greatest interest for many readers will be how these compare, even whilst accepting that the cohort size means that the chances of there being a significant difference between the two is remote. It may be that the authors are dissuaded from making this comparison because of the lack of a true difference between the treatment and non-treatment groups. However, this is still important, and there are many possible explanations which would not negate the case for doing genomic analyses.

The swimmer plot in Figure 5 shows that there are sufficient therapy responses to suggest that genomic-informed therapy selections can result in meaningful clinical impacts for individual patients. In this figure (Figure 5 B) the therapies that were instituted in each individual are not shown. Adding this information would add greatly to this figure.

The tumour immunological analysis presented in Figure 4, relating this feature to mutation burden, is a relatively unique aspect of this report, and one that could be strengthened which would add to the impact of the work. This analysis is restricted to algorithmically-predicted CD8+ T-cells scores (using CIBERSORT). However, there are many other potential features which CIBERSORT (or other tools) predict which may interest readers. For example, are there associations with myeloid suppressor cells, or other T cell subsets? Is there any association between TMB and expression of checkpoints (such as PD-L1 or CTLA4). What was the checkpoint expression in the patient who responded to ICI therapy? It is noticeable in 4C and 4D that the "Other Solid" group has samples which have a higher CD8 and total T cell scores. It would interest readers to know what these tumours were or what their driver mutations (or germline) mutations were.

In Figure 3, the authors present interesting data about the range of variants that are potentially actionable. I think there are some elements of the figure, especially 3A that the authors might consider improving. For example, in the Cell Cycle genes, all have alterations in gene expression and copy number, but one imagines that for CDK4 and 6 this will be gene amplification and high expression, whereas for CDKN2A and B, these will be Gene deletions and low expression. However the figure makes no distinction between copy number gains or losses and gene expression be elevated or lost. Further, where gene expression is the sole targetable aberration, is this elevated gene expression?

In the text, the authors described 2 individuals in which high-level tiered fusions had not been identified prior to enrolment in POG. The reference is to Figure 3B but this really doesn't describe these individuals or why these fusions might have been missed or how they were identified in the sequencing study. This

perhaps deserves a little more description about their discovery and perhaps what the consequences were.

Reviewer #3 (Remarks to the Author): Expert in cancer genomics and transcriptomics, clinical and computational genomics, and paediatric cancers

In the present study Deyell and colleagues describe the personalized oncogenomics program (POG) experience, which has profiled >1.2K patients by comprehensive genome profiling, for a subset of 91 pediatric cases. Of these, 88 consented to participated and WGS analysis (somatic and germline) and RNA-seq was successful for 89.7% (n=78). The cohort included 46 patients with solid non- CNS tumors, 19 CNS, 6 hematological neoplasms and 8 benign cases including subjects with aggressive fibromatosis or plexiform neurofibromas. This makes the total tumor dataset n=71.

POG workflow

The focus of this manuscript is on the findings by WGTS. It would be useful to provide a description of what other molecular testing these individuals have received clinically. This would allow the reader to put into context any novel (previously unappreciated findings) of the POG program (e.g BCR:ABL and CCDC6:RET fusions). In the absence of comparator data, it is also challenging to assess the sensitivity and specificity of the POG workflow.

Partial findings (targeted analysis of WGTS data) were returned at a median time of 34 days from sample receipt. Full WGTA report was available at a median of 71 days which is 2+ months. For 1 in 4 participants to the study the report was generated after the participant had expired. The authors should provide context on how this compares to other pediatric WGTS initiatives as the turn around time frames here are much longer to the ones (~2 weeks) reported in the literature.

Reporting of WGTS findings

The overall distribution of germline findings was consistent with prior findings. The same was true for somatic mutations. There is very little detail provided on the process for reporting and validating germline findings. Reporting of SV is restricted to those with concomitant evidence by RNA for the formation of established oncogenic fusion genes. What about SVs that may be clinically relevant but that do not result in fusion genes? Mutation signature analysis offers an opportunity to validate mutations in MMR/DDR genes, however the authors do not include such a process in their methods. Would be helpful to commend whether this was evaluated in their workflow and mutation signature analysis was considered for any germline VUS mutations.

There is very little information on the results for the 6 patients with hematological neoplasms. If no notable results perhaps this study can focus on the solid tumors / benign tumors.

Among 79 participants, 73 (92%) had at least one expression-based alteration. The majority of these were low or moderate evidence to support actionability. The authors mention that of 171 therapeutically actionable alterations, 41% were supported by DNA-level evidence, 39% were supported by RNA expression evidence, and 20% were supported by a combination. In Figure 3 it would be helpful to incorporate a direction based color scheme for aberrant expression (downregulated or upregulated). In the text and methods it would be helpful if the authors could add a bit more information on how DNA alone, or in combination to RNA was used to render further support into the putative expression based biomarkers. Some specific examples pointing to the precise actionability of those putative biomarkers and if that is within the context of the tumor type in which it was found would also be important to incorporate. The authors could consider accompanying the paragraph between lines 154-162 with a supplemental table outlining each of the cases, tumor type, genomic alteration, proposed actionable therapy, levels of evidence supporting the therapy and if that therapy was available in that tumor type with recorded response status. This would also be relevant as supportive evidence for the results section outlined in Figure 5. This part of the results forms the most important part of this study and would be of interest to pediatric oncology.

The authors describe an evaluation of tumor immune cell infiltrates for a subset of the cases. They identify no correlation between TMB and immune cell infiltration. The authors propose an association between specific alterations (i.e. SMARCB1) or tumor types (rhabdoid/chordomas) with high tumor cell infiltrates. They offer no detail on subtypes (e.g in neuroblastoma) or whether any of these observations can be extended or are validated in other published pediatric or AYA cohorts (Sick Kids, DKFZ, MSKCC). Did the authors evaluate coding TMB as well as genomewide TMB?

There is limited information on what proportion of the findings would be captured by DNA or RNA alone-seq alone vs a combination.

Discussion

I would encourage the authors to reword the first paragraph to 96% had putative or potential therapeutically actionable events. The current wording is too strong for the level of evidence provided. There is very limited discussion on how the workflow, findings and novel insights of this study compares to the existing literature (turn around timeframe, success rate, actionability, tumor types, outcomes) beyond the comparisons provided to the childhood zero programs, nor what are the novel insights that emerge from this study relative to the prior literature.

Overall, whilst the study is of interest and very important there is not much detail in the methods nor results of this manuscript, and limited novelty relative to the published literature.

REVIEWER COMMENTS AND RESPONSES

Reviewer #1: Expert in paediatric cancer clinical research and genomics

The authors report on results from the Pediatric Personalized OncoGenomic cohort, the adult counterpart was reported in 2020. Following consent, tumor and matched normal tissue underwent whole genome and transcriptome profiling. Pediatric cancer genomic profiling by next generation sequencing has been previously published as referenced by the authors. However, there are some elements in this program that have not been a focus of prior pediatric efforts. Specifically, gene expression profiles and immune profiles. Overall the study is well done with interesting results.

- 1. Cohort is relatively small with a paucity of hematologic malignancies despite the high frequency of these malignancies in pediatrics which is a major weakness of the manuscript.*

Response: We acknowledge and appreciate the reviewer's point; this is a heterogeneous pediatric cohort including predominantly solid and CNS tumors with enrollment starting in 2013. Although patients with hematologic malignancies were eligible, there were challenges to their inclusion related to the need for rapid turnaround times and for adequate tumor content from bone marrow or CSF samples for genomic sequencing. In our successor cohort, and national Canadian PROFYLE study, we have prioritized inclusion of hematologic malignancies with consideration of upfront next generation sequencing.

This limitation is acknowledged in our Discussion: "Hematologic malignancies were underrepresented due to difficulties acquiring suitable samples with adequate tumor content, rapidity of disease progression, and availability of alternative therapeutic strategies."

-
- 2. Given prior pediatric studies did not include gene expression in their studies but rather focused on somatic and germline SNV/Indel/CNA/SV, I would recommend expanding this component of the manuscript. The clinical outcomes description is described based on LOE, however gene expression is included in LOE3-5. What was the contribution of expression signatures to patients receiving targeted therapy and the response? In many cases, the somatic mutations/fusions dictate the gene expression. Were there cases where expression signatures were found in the absence of a pathogenic mutation in a given pathway leading to use of a targeted agent and clinical response? This would*

argue that expression signatures are more than just a complement to the detection of mutations.

Response: We're pleased that there is interest in expression data and signatures as we feel it is a crucial part of WGTA and contributes beyond what can be captured by DNA sequencing alone. As stated in the text, 38% of therapeutically actionable findings were based on RNA evidence alone, which includes both immune profiling scores and outlier expression of individual genes. However, many of these were not acted on by a clinician, likely because of the uncertainty around expression signatures as supporting evidence at the time. We included an example of a patient with chordoma that received an immune checkpoint inhibitor, based on an outlier T-cell score and high expression of brachyury (TBXT), and who achieved a radiographic partial response. Other patients achieved prolonged stable disease (≥ 6 mos) with a VEGFR inhibitor, based on high *VEGFA* expression, and irbesartan, based on high *FOS/JUN* expression. We have included a new Supplementary Table (S4) that includes the treatments received by all patients, the target gene, target category (DNA, RNA, combined), as well as the radiographic response to more clearly describe this. We currently use expression analyses to determine activation of a molecular pathway, and this is incorporated into our molecular tumor board (MTB) report as supportive evidence. We have added an example pathway diagram as our new Supplementary Fig. 2 to demonstrate to readers how we integrate data from DNA and RNA to provide a more comprehensive understanding of an individual cancer's drivers for MTB discussions.

We have added the following paragraph to our Results section: "Among 32 therapies pursued, 11 were based on RNA evidence only and another 14 were supported by combined DNA and RNA alterations (Supplementary Table S4). Forty-four percent ($n=4/9$) of patients assessed for response to therapies based on RNA only evidence achieved benefit (PR or prolonged SD), while 50% ($n=6/12$) benefited from therapies based on combined DNA/RNA evidence."

We also explored survival outcomes by molecular target category (RNA versus DNA/combined molecular target; see below) for patients who received targeted therapy, as requested. However, we are concerned that small numbers ($n=28$ received an informed therapy), heterogeneity of tumor types, and selection bias make it impossible to confidently draw conclusions. Patients with locally advanced, benign tumors received therapy based on RNA targets, but likely have superior survival outcomes based purely on the underlying biology of their disease. Patients were often considered for lower level evidence therapy suggestions if they had few alternatives, but more indolent disease. We have included two survival analyses for the Reviewer here by molecular target type in patients with solid tumors only, excluding benign tumors to reduce heterogeneity, but feel we have insufficient data to conclusively answer this question.

Top: Overall survival for patients with solid tumors only for first molecularly informed therapy (n=19), by data type (RNA, DNA or both). Hazard ratio for RNA vs. DNA (reference) is shown on the figure. **Bottom:** Overall survival for patients with solid tumors only (no benign/CNS), by RNA evidence alone vs. DNA alone/combination.

- 3. Supplementary Table targeted therapies would benefit from adding the molecular mutation/gene expression signature that led to the recommended targeted drug as well as the response. Currently the table is limited to the level of evidence and there is no response information.*

Response: We agree that this information may be useful to the reader and have updated Supplementary Table S4 to include the molecular targets and radiographic response.

- 4. In the last section of results, 53.3% of the 28 patients that pursued molecularly targeted therapies had a PR/CR/SD. The discussion specifies 48% of patients benefitted. Please clarify this discrepancy.*

Response: The discrepancy here was due to exclusion of patients with locally advanced, unresectable benign tumors, bringing the values down from 15/28 (53.5%) to 12/25 (48%). We have clarified this in the Results section as follows: "Among the 28 therapy trials assessed for response, five achieved radiographic response (partial response (PR) n=2, complete response (CR) n=3) and ten had stable disease (SD) for ≥ 6 months for an overall benefit rate of 54%. If patients with locally advanced, benign tumors were excluded (n=3), the overall benefit rate was 48%."

Reviewer #2: Expert in paediatric cancer clinical research, genomics, and tumour immune microenvironment

The manuscript from Deyell and colleagues describes the whole genome and transcriptome analysis of paediatric patients enrolled in the POG study between 2013 and 2019 in British Columbia in Canada. In this manuscript the authors report on the 79 participants who received results from this study. The primary strength of the work is that it is one of only a few reports of genomics analysis being applied across all cancer subtypes (so-called “pan-cancer”) and of genomics being applied in a clinical setting, that is with a primary purpose of informing clinical decision making. In this context, this manuscript is an important contribution to the bodies of work assessing the clinical impacts of genomics in paediatric cancer.

- 1. The weaknesses of the manuscript are the relatively small cohort size (which is not changeable and which is a relatively minor issue for this reviewer) and the very descriptive style of manuscript (which is changeable and more important). Whilst it is not this reviewer’s intention to impose any sort of writing style on the authors, one suspects that plain descriptions of the cohort, and range of mutations and other features are likely to be less interesting to readers (because things like this have already been published) than elements of the work which expand on the potential utility of genomic analysis of cancers (for example the immunoprofiling), ways in which different data types (WGS and RNA) are operationally integrated in developing patient reports and measures of the potential impact of this program. For example, In Figure 1D and E the authors show the Kaplan-Meier curves for the cohort dependent on the diagnostic category and in Figure 5 C-F they show the curves for the POG-therapy treated patients. One imagines that the greatest interest for many readers will be how these compare, even whilst accepting that the cohort size means that the chances of there being a significant difference between the two is remote. It may be that the authors are dissuaded from making this comparison because of the lack of a true difference between the treatment and non-treatment groups. However, this is still important, and there are many possible explanations which would not negate the case for doing genomic analyses.*

Response: We agree that much of the novelty of our cohort lies in our incorporation of RNA-based data to support therapeutic choices in the clinic. To expand on the immunoprofiling aspect of this work, we have included a heatmap in Supplementary Fig. 1 showing all CIBERSORT cell types across this pan-cancer cohort. This analysis further emphasizes that the different cell types are found across many samples, regardless of tumor type or biopsy site.

We have further expanded our description of the real-time, operational integration of both DNA and RNA-sequencing results for all molecular tumor board (MTB) reports in the Methods section: “Genomic data (SNV, CNV, SV) and transcriptomic data (SV, gene expression) are integrated at both gene level and pathway level. Genes that directly or indirectly interact in the same pathway are analyzed as a unit and visualized in a pathway plot (Supplementary Fig. 2). Potential cause-effect links are established between alterations in upstream regulators and expression level of downstream effectors, which are integrated to identify pathway level aberrations and therapeutic targets.”

As is shown in our additional Supplementary Table S4, RNA-based expression outlier status was used as both supportive evidence of molecular pathway activation in presence of additional DNA alterations, or in some cases without concurrent DNA-based alterations (typically at lower levels of evidence). The incorporation of transcriptome-informed data into therapeutic actionability is novel and we show that some patients achieved clinical benefit. As noted above, we have added an example pathway diagram (Supplementary Fig. 2) to demonstrate how we integrated DNA and RNA data to provide a comprehensive understanding of an individual cancer’s drivers for MTB discussions. We have added text in Results to outline the proportion of therapeutically actionable variants that were identified by DNA, combined DNA/RNA or RNA alone. In addition, we have highlighted which patients received targeted therapies based on evidence from RNA-based expression, along with their responses.

We agree with the reviewer that determining the clinical benefit of WGTA is of critical importance moving forward, however, as the reviewer states, there are difficulties in doing so in this cohort. The small sample size and heterogeneity of tumor types play a role. There is also likely significant selection bias in that those patients receiving molecularly informed therapies in our cohort may have had more indolent disease allowing them time to consider and access a novel therapy. Patients with rapidly progressive disease or declining clinical status may be less likely to be able to access additional therapy. As such, we can predict that patients who accessed therapies may have superior survival outcomes purely related to the underlying differences in disease biology (and unrelated to therapy response). In addition, when assessing survival outcomes for patients who received POG-informed therapy, we assessed survival from the date of treatment initiation. However, in order to compare with the remainder of the (untreated) cohort we would need to define time zero at study entry/sample receipt and this may further exacerbate biases. Future, prospective clinical trials are required to adequately assess overall clinical benefit of molecularly informed therapies.

- 2. The swimmer plot in Figure 5 shows that there are sufficient therapy responses to suggest that genomic-informed therapy selections can result in meaningful clinical impacts for individual patients. In this figure (Figure 5B) the therapies that were instituted in each individual are not shown. Adding this information would add greatly to this figure.*

Response: The therapies and drug class received by each patient, in addition to their radiographic response to that therapy, have been added to Supplementary Table S4. In addition, we have added an annotation for drug category in the swimmer plot in Fig. 5B (now Fig. 6B).

- 3. The tumour immunological analysis presented in Figure 4, relating this feature to mutation burden, is a relatively unique aspect of this report, and one that could be strengthened which would add to the impact of the work. This analysis is restricted to algorithmically-predicted CD8+ T-cells scores (using CIBERSORT). However, there are many other potential features which CIBERSORT (or other tools) predict which may interest readers. For example, are there associations with myeloid suppressor cells, or other T cell subsets? Is there any association between TMB and expression of checkpoints (such as PD-L1 or CTLA4). What was the checkpoint expression in the patient who responded to ICI therapy? It is noticeable in 4C and 4D that the “Other Solid” group has samples which have a higher CD8 and total T cell scores. It would interest readers to know what these tumours were or what their driver mutations (or germline) mutations were.*

Response: We thank the reviewer for highlighting other areas that may be of interest to readers. The outliers in the “other solid group” are two chordomas and a rhabdoid tumor. We have annotated these in Fig. 4C/D (now Fig. 5C/D) to make this clearer to the reader. The driver alterations for these tumor types are reported in the text: they all share loss or inactivation of SMARCB1. The patient who responded to ICI therapy had a chordoma and has been previously reported in Williamson et al., (2021), referenced in the text of the Results section. We have expanded the text to note the high expression of PD-L1 and the validation of immune cell scores with IHC.

CD8+ and total T-cells were primarily included in this study as these were cell types of most interest that were discussed at MTB meetings. To expand the cell types in this study, we have included a heatmap in the Supplementary Figures (Supplementary Fig. 1) of all the cell types from CIBERSORT which further demonstrates that there is no bias by primary tumor type. We

also added in the manuscript the lack of correlation between TMB and expression of checkpoint genes: “Similarly, no strong correlation was observed between TMB and expression of several immune checkpoint genes, including CTLA4 ($r=0.14$, $P=0.22$, Spearman), PD-L1 ($r=-0.24$, $P=0.04$), PD-1 ($r=0.15$, $P=0.21$).”

4. *In Figure 3, the authors present interesting data about the range of variants that are potentially actionable. I think there are some elements of the figure, especially 3A that the authors might consider improving. For example, in the Cell Cycle genes, all have alterations in gene expression and copy number, but one imagines that for CDK4 and 6 this will be gene amplification and high expression, whereas for CDKN2A and B, these will be Gene deletions and low expression. However the figure makes no distinction between copy number gains or losses and gene expression be elevated or lost. Further, where gene expression is the sole targetable aberration, is this elevated gene expression?*

Response: We thank the reviewer for this point of clarification. We have revised Fig. 3 and now differentiate both the copy number variants and expression outliers into high or low categories.

5. *In the text, the authors described 2 individuals in which high-level tiered fusions had not been identified prior to enrolment in POG. The reference is to Figure 3B but this really doesn't describe these individuals or why these fusions might have been missed or how they were identified in the sequencing study. This perhaps deserves a little more description about their discovery and perhaps what the consequences were.*

Response: We agree with the reviewer that these two individuals were very interesting given that the research sequencing identified actionable fusions that were not identified in routine clinical care. The patient who had a POG-identified BCR:ABL fusion initially presented with a distal humerus mass. A biopsy was consistent with lymphoma and all routine clinical testing was performed, including FISH for BCR:ABL which was negative. This patient therefore was treated for B-lymphoblastic lymphoma and did not receive TKI therapy that would have been included if the fusion had been identified clinically. The patient unfortunately relapsed, enrolled in this study, BCR:ABL was identified and was subsequently offered salvage chemotherapy with concurrent imatinib, which was declined by the patient and family. This case highlights the possibility that despite routine clinical testing, actionable variants can be missed.

The second patient had papillary thyroid carcinoma who relapsed in 2016, a time at which routine genomic testing was not being pursued locally. They were enrolled in POG to identify any recurrent, potentially actionable targets and a CCDC6:RET fusion was identified. Despite a high prevalence of actionable findings in refractory pediatric thyroid carcinoma, routine genomic testing was not yet standard of care, though this has now evolved. This study patient has subsequently remained clinically well with no evidence of disease recurrence.

We have added the following to the Results: “Two participants with high-level therapeutically actionable variants (LOE1-2) identified through POG (prior clinical fluorescence in situ hybridization test was negative for BCR:ABL in lymphoma; no prior test was done for CCDC6:RET in papillary thyroid carcinoma) did not receive targeted therapy due to patient choice and no evidence of disease, respectively.”

Reviewer #3: Expert in cancer genomics and transcriptomics, clinical and computational genomics, and paediatric cancers

In the present study Deyell and colleagues describe the personalized oncogenomics program (POG) experience, which has profiled >1.2K patients by comprehensive genome profiling, for a subset of 91 pediatric cases. Of these, 88 consented to participate and WGS analysis (somatic and germline) and RNA-seq was successful for 89.7% (n=78). The cohort included 46 patients with solid non- CNS tumors, 19 CNS, 6 hematological neoplasms and 8 benign cases including subjects with aggressive fibromatosis or plexiform neurofibromas. This makes the total tumor dataset n=71.

- 1. The focus of this manuscript is on the findings by WGTS. It would be useful to provide a description of what other molecular testing these individuals have received clinically. This would allow the reader to put into context any novel (previously unappreciated findings) of the POG program (e.g BCR:ABL and CCDC6:RET fusions). In the absence of comparator data, it is also challenging to assess the sensitivity and specificity of the POG workflow.*

Response: All study participants were pediatric oncology patients at our provincial quaternary care pediatric oncology clinical program in Western Canada and received standard of care clinical genomic testing based on their tumor type at time of cancer diagnosis. Patients were enrolled into the study from 2013 to 2019, and routine clinical molecular diagnostic testing has evolved over that time period. The two previously unrecognized high-tier fusions in our cohort have been further described in response to Reviewer 2 above and in the text. Despite routine BCR:ABL clinical testing for lymphoma, this patient's clinical test result was negative at diagnosis, highlighting the possibility of missed high-level variants in clinical care. The subsequent CCDC6:RET fusion was identified in a papillary thyroid carcinoma (PTC) patient at relapse prior to routine clinical molecular characterization of pediatric PTC at our site.

-
- 2. Partial findings (targeted analysis of WGTS data) were returned at a median time of 34 days from sample receipt. Full WGTA report was available at a median of 71 days which is 2+ months. For 1 in 4 participants to the study the report was generated after the participant had expired. The authors should provide context on how this compares to other pediatric WGTS initiatives as the turnaround time frames here are much longer to the ones (~2 weeks) reported in the literature.*

Response: Our pediatric cohort began enrolling in 2013, at which time the technical and analytical processes for high through-put WGTA had not yet been optimized. In similar years, reported turnaround times (TATs) for studies undertaking comprehensive WGTA range from 3-8 weeks (Mody et al., 2015; Newman et al., 2021).

Our TAT for WGTA (sequencing and bioinformatics analyses) was calculated from time of sample receipt to date of molecular tumor board (MTB) discussion, which was scheduled after the final genomic report was signed off and shared. Samples were typically batched for sequencing and patients who died prior to report completion were often deprioritized for bioinformatic analyses and case presentation.

Our whole genome sequencing was completed, and a targeted report was released in a median of 34 days, which required little manual bioinformatic analysis time. The additional time to full MTB was utilized to complete comprehensive bioinformatics analyses, including RNA-sequencing pathway diagrams for each tumor (now included as Supplementary Fig. 2).

For other pediatric studies undertaking comprehensive WGTS and analysis, TATs are improving over time. The St. Jude pediatric cohort (recruited 2015-2017) had a TAT of <7 weeks in 95% (Newman et al., 2021). More recently, the INFORM pediatric cohort (whole exome, low coverage WGS and RNA) reported a median TAT of 25 days. The MSKCC group developed a workflow to report integrated WGTA results in <2 weeks, though this appears to be an automated pipeline and reporting as opposed to additional manual bioinformatic analytic time (Shukla et al., 2022; van Tilburg et al., 2021).

We have added this to the Discussion of limitations for our cohort: “A move towards an increasingly streamlined WGTA workflow will further optimize result utility and is evident in recent pediatric precision oncology cohorts with reported TATs of <2 weeks¹⁶.”

3. Reporting of WGTS findings

The overall distribution of germline findings was consistent with prior findings. The same was true for somatic mutations. There is very little detail provided on the process for reporting and validating germline findings. Reporting of SV is restricted to those with concomitant evidence by RNA for the formation of established oncogenic fusion genes. What about SVs that may be clinically relevant but that do not result in fusion genes? Mutation signature analysis offers an opportunity to validate mutations in MMR/DDR genes, however the authors do not include such a process in their methods. Would be

helpful to commend whether this was evaluated in their workflow and mutation signature analysis was considered for any germline VUS mutations.

Response: Additional detail has been provided in Methods on the process of reporting and validating germline findings.

We thank the reviewer for bringing up the discussion on how structural variants (SVs) have been used in the analysis. In addition to looking at fusions of known oncogenic or targetable genes, we also calculate a genome-wide SV burden to evaluate genome stability for each patient, which includes SVs in intergenic regions and those only observed in DNA. In addition, SVs are used in the calculation of HRD score to assess for homologous recombination deficiency (HRD). In this pediatric cohort, we did not observe high HRD scores as we did in the adult cohort of POG, which is likely due to the lack of mutations in BRCA1/2 or other HR genes in these disease types.

We added the following paragraph to the Results to describe the use of SV in our analysis: “The SV burden was calculated for each sample to evaluate genome stability. SV is also used to calculate HRD score²⁶ to detect deficiency in homologous recombination. The HRD scores in this cohort ranged from 0-45, which corresponds to the 0-92 (median 23) percentile when compared to the adult cohort of POG. High HRD scores were observed in osteosarcoma, in which chromosomal rearrangements are often observed²⁷.”

Of note, we detected a pathogenic *PALB2* germline variant which was not associated with somatic signatures of HRD in a patient with glioblastoma. This has been added to the Results section.

4. There is very little information on the results for the 6 patients with hematological neoplasms. If no notable results perhaps this study can focus on the solid tumors / benign tumors.

Response: We agree that hematological neoplasms in this study are under-represented in our study, along with other pediatric cohorts (<10% of INFORM study patients)(van Tilburg et al., 2021). Among six patients with five diagnoses, one had a *BCR:ABL* fusion, two acute lymphoblastic leukemia patients had copy number alterations in *CDKN2A*, and none had pathogenic/likely pathogenic germline alterations.

We acknowledge the small number of hematologic malignancies in our limitation section but would like to include them in our results to keep a representative view of pediatric patients seen in a clinical program such as ours.

- 5. Among 79 participants, 73 (92%) had at least one expression-based alteration. The majority of these were low or moderate evidence to support actionability. The authors mention that of 171 therapeutically actionable alterations, 41% were supported by DNA-level evidence, 39% were supported by RNA expression evidence, and 20% were supported by a combination. In Figure 3 it would be helpful to incorporate a direction based color scheme for aberrant expression (downregulated or upregulated). In the text and methods it would be helpful if the authors could add a bit more information on how DNA alone, or in combination to RNA was used to render further support into the putative expression based biomarkers. Some specific examples pointing to the precise actionability of those putative biomarkers and if that is within the context of the tumor type in which it was found would also be important to incorporate.*

Response: We agree with this suggestion and have differentiated expression outliers (high or low) and copy number alterations (gains or losses) in revised Fig. 3A.

We also agree that key to the novelty of our report is our integration of RNA-based evidence into therapeutic actionability. We have expanded our description of the real-time, operational integration of both DNA and RNA-sequencing results for all molecular tumor board reports (MTBs) in Methods and have included a new Supplementary Fig. 2 which includes an example pathway diagram demonstrating how RNA is integrated into a complete analysis presented at the MTB meeting.

We have added expanded text and added to Supplementary Table S4 to describe molecularly informed therapy trials that were supported by combined DNA/RNA evidence or RNA evidence alone. Beyond the limited number of therapy trials pursued by participants, we have included an additional Supplementary Table S3 with all putative actionable variants, annotated by variant category and level of evidence.

- 6. The authors could consider accompanying the paragraph between lines 154-162 with a supplemental table outlining each of the cases, tumor type, genomic alteration, proposed actionable therapy, levels of evidence supporting the therapy and if that therapy was available in that tumor type with recorded response status. This would also*

be relevant as supportive evidence for the results section outlined in Figure 5. This part of the results forms the most important part of this study and would be of interest to pediatric oncology.

Response: We thank the reviewer for the suggestion of including a new Supplementary Table with each case, genomic alteration, and actionable therapy. We have provided this in Supplementary Tables S3 (all cases) and S4 (molecularly informed therapies pursued).

7. The authors describe an evaluation of tumor immune cell infiltrates for a subset of the cases. They identify no correlation between TMB and immune cell infiltration. The authors propose an association between specific alterations (i.e. SMARCB1) or tumor types (rhabdoid/chordomas) with high tumor cell infiltrates. They offer no detail on subtypes (e.g in neuroblastoma) or whether any of these observations can be extended or are validated in other published pediatric or AYA cohorts (Sick Kids, DKFZ, MSKCC). Did the authors evaluate coding TMB as well as genomewide TMB?

Response: We agree with the reviewer that these associations warrant further exploration in other cohorts as there may be clinical utility for immune checkpoint inhibitor (ICI) therapy in patients with certain genomic features, such as loss of *SMARCB1*. We have previously demonstrated that two of the chordoma patients from our study had high immune scores in comparison with external datasets including Treehouse, TARGET, and a published cohort of rhabdoid tumors (Williamson et al., 2021). The high T-cell scores in these patients were also validated by IHC. We have expanded the text around this to include this comparison with other datasets. Cytotoxic T-cell infiltration has been previously described in malignant rhabdoid tumors (both extra-cranial and atypical teratoid rhabdoid tumors) which are also driven by *SMARCB1* loss (Chun et al., 2019). Interestingly, rare exceptional responders to ICI monotherapy in non-TMB-high unselected pediatric tumors have also been reported in patients with malignant rhabdoid tumors, epithelioid sarcoma and chordomas, which typically have low TMB, outlier high immune infiltration and mutations affecting the SWI/SNF complex (*SMARCB1/A4*) (Long et al., 2022).

We have four patients with inferred loss of function *SMARCB1* events (2 CNVs, loss of function mutation, and low expression), and found a trend between these events and CD8+ T-cell scores, however, the low numbers meant this did not achieve statistical significance (p=0.19, figure below).

We also observe CD8+ and total T-cell immune infiltration outliers among other tumor types including neuroblastoma, soft tissue sarcoma and high-grade CNS tumors (Fig 5C/D). Immunotranscriptomic analyses in large cohorts of pediatric solid tumors including neuroblastoma and sarcoma have demonstrated subsets with prominent CD8+ T-cell infiltration, correlation of immunomodulatory gene expression with CD8+ T-cell scores and potential impact on survival outcomes (Brohl et al., 2021). In neuroblastoma, tumor infiltrating lymphocytes and expression of checkpoint genes have been associated with *MCYN-NA* tumors. We have added these points to our Discussion.

For TMB, we did initially evaluate both coding and genomic TMB but found them to be highly correlated ($R=0.95$, $p<2.2\times 10^{-16}$, Spearman), and therefore exhibited the same trends with immune cell correlation. We have included a statement in the manuscript on the coding TMB.

8. *There is limited information on what proportion of the findings would be captured by DNA or RNA alone-seq alone vs a combination.*

Response: We have added this for all cases and specifically for molecularly informed therapy trials in Supplementary Tables S3 and S4. We have also added text in Results outlining the proportion of therapeutically actionable targets informed by RNA sequencing alone, or combined with DNA, overall and specifically for those patients who received molecularly informed therapy, as noted above.

9. *Discussion*

I would encourage the authors to reword the first paragraph to 96% had putative or potential therapeutically actionable events. The current wording is too strong for the level of evidence provided.

Response: Agreed and revised.

10. There is very limited discussion on how the workflow, findings and novel insights of this study compares to the existing literature (turn around timeframe, success rate, actionability, tumor types, outcomes) beyond the comparisons provided to the childhood zero programs, nor what are the novel insights that emerge from this study relative to the prior literature.

Response: We thank the reviewer for this comment and have expanded the Methods section to better outline workflow and integration of RNA sequencing analyses prior to molecular tumor board reports. We have expanded our Discussion section to include more comparison to other large pediatric advanced cancer cohorts, including MSKCC, INFORM at DKFZ, and the KICS Canadian cohort, in addition to the Australian ZERO cohort. We also acknowledge timelines in our limitations, and though this continues to improve over time, we note that the routine incorporation of transcriptome data into analyses did extend turnaround times beyond our more automated “targeted gene report” based on WGS.

As our reviewer notes, our germline and somatic DNA-based alterations are similar to previously described pediatric cohorts. We have expanded on the novelty of this cohort which includes the real-time integration of RNA-sequencing data as supportive, and sometimes stand-alone evidence for therapeutic actionability. Unlike prior cohorts, some of which have included transcriptome sequencing, we have now clearly delineated all actionable targets by both “level of evidence” as well as molecular data source (DNA, RNA, combined) (Tables S3 and S4). We acknowledge that there are challenges to the routine incorporation of transcriptome-level evidence, but we highlight its potential utility in delineating molecular pathway activation, elucidation of underlying diagnosis and drivers inferred by expression signatures, and an improved understanding of tumor immune infiltration in a time when more immune-based therapies are explored in pediatric oncology. We also have the advantage of being able to directly compare our pediatric cancer cohort to our large adult, advanced pan-cancer cohort that utilized the same workflow and WGTA pipeline.

11. Overall, whilst the study is of interest and very important there is not much detail in the methods nor results of this manuscript, and limited novelty relative to the published literature.

Response: We thank the reviewer for their interest and have shifted the focus away from DNA-based actionable findings in both tumor and germline, which correlate well with prior pediatric reports, to focus on the novelty of the integration of transcriptome-level support for therapeutic actionability, along with description of the molecularly informed therapies pursued subsequent to study participation in the context of real-world clinical care. We have further expanded our Methods section to describe this integration, have expanded our Results section to include all putative actionable variants (annotated by level of evidence and target category), and our Discussion section to highlight the potential clinical utility of our approach in pediatric patients with poor prognosis cancers.

REFERENCES

- Brohl, A. S., Sindiri, S., Wei, J. S., Milewski, D., Chou, H.-C., Song, Y. K., Wen, X., Kumar, J., Reardon, H. V., Mudunuri, U. S., Collins, J. R., Nagaraj, S., Gangalapudi, V., Tyagi, M., Zhu, Y. J., Masih, K. E., Yohe, M. E., Shern, J. F., Qi, Y., ... Khan, J. (2021). Immuno-transcriptomic profiling of extracranial pediatric solid malignancies. *Cell Reports*, *37*(8), 110047. <https://doi.org/10.1016/j.celrep.2021.110047>
- Chun, H.-J. E., Johann, P. D., Milne, K., Zapatka, M., Buellbach, A., Ishaque, N., Iskar, M., Erkek, S., Wei, L., Tessier-Cloutier, B., Lever, J., Titmuss, E., Topham, J. T., Bowlby, R., Chuah, E., Mungall, K. L., Ma, Y., Mungall, A. J., Moore, R. A., ... Kool, M. (2019). Identification and analyses of extra-cranial and cranial rhabdoid tumor molecular subgroups reveal tumors with cytotoxic t cell infiltration. *Cell Reports*, *29*(8), 2338-2354.e7. <https://doi.org/10.1016/j.celrep.2019.10.013>
- Long, A. H., Morgenstern, D. A., Leruste, A., Bourdeaut, F., & Davis, K. L. (2022). Checkpoint immunotherapy in pediatrics: Here, gone, and back again. *American Society of Clinical Oncology Educational Book. American Society of Clinical Oncology. Annual Meeting*, *42*, 1–14. https://doi.org/10.1200/EDBK_349799
- Mody, R. J., Wu, Y.-M., Lonigro, R. J., Cao, X., Roychowdhury, S., Vats, P., Frank, K. M., Prensner, J. R., Asangani, I., Palanisamy, N., Dillman, J. R., Rabah, R. M., Kunju, L. P., Everett, J., Raymond, V. M., Ning, Y., Su, F., Wang, R., Stoffel, E. M., ... Chinnaiyan, A. M. (2015). Integrative clinical sequencing in the management of refractory or relapsed cancer in youth. *JAMA*, *314*(9), 913–925. <https://doi.org/10.1001/jama.2015.10080>
- Newman, S., Nakitandwe, J., Kesserwan, C. A., Azzato, E. M., Wheeler, D. A., Rusch, M., Shurtleff, S., Hedges, D. J., Hamilton, K. V., Foy, S. G., Edmonson, M. N., Thrasher, A., Bahrami, A., Orr, B. A., Klco, J. M., Gu, J., Harrison, L. W., Wang, L., Clay, M. R., ... Nichols, K. E. (2021). Genomes for kids: The scope of pathogenic mutations in pediatric cancer revealed by comprehensive DNA and RNA sequencing. *Cancer Discovery*, *11*(12), 3008–3027. <https://doi.org/10.1158/2159-8290.CD-20-1631>
- Shukla, N., Levine, M. F., Gundem, G., Domenico, D., Spitzer, B., Bouvier, N., Arango-Ossa, J. E., Glodzik, D., Medina-Martínez, J. S., Bhanot, U., Gutiérrez-Abril, J., Zhou, Y., Fiala, E., Stockfisch, E., Li, S., Rodriguez-Sanchez, M. I., O'Donohue, T., Cobbs, C., Roehrl, M. H. A., ... Papaemmanuil, E. (2022). Feasibility of whole genome and transcriptome profiling in pediatric and young adult cancers. *Nature Communications*, *13*(1), 2485. <https://doi.org/10.1038/s41467-022-30233-7>

van Tilburg, C. M., Pfaff, E., Pajtler, K. W., Langenberg, K. P. S., Fiesel, P., Jones, B. C., Balasubramanian, G. P., Stark, S., Johann, P. D., Blattner-Johnson, M., Schramm, K., Dikow, N., Hirsch, S., Sutter, C., Grund, K., von Stackelberg, A., Kulozik, A. E., Lissat, A., Borkhardt, A., ... Witt, O. (2021). The pediatric precision oncology INFORM registry: Clinical outcome and benefit for patients with very high-evidence targets. *Cancer Discovery*, *11*(11), 2764–2779. <https://doi.org/10.1158/2159-8290.CD-21-0094>

Williamson, L. M., Rive, C. M., Di Francesco, D., Titmuss, E., Chun, H.-J. E., Brown, S. D., Milne, K., Pleasance, E., Lee, A. F., Yip, S., Rosenbaum, D. G., Hasselblatt, M., Johann, P. D., Kool, M., Harvey, M., Dix, D., Renouf, D. J., Holt, R. A., Nelson, B. H., ... Marra, M. A. (2021). Clinical response to nivolumab in an INI1-deficient pediatric chordoma correlates with immunogenic recognition of brachyury. *NPJ Precision Oncology*, *5*(1), 103. <https://doi.org/10.1038/s41698-021-00238-4>

REVIEWERS' COMMENTS

Reviewer #1 (Remarks to the Author):

I am satisfied by the authors responses to the reviewer comments. The clarifications and additions to the manuscript have strengthened it.

Reviewer #2 (Remarks to the Author):

The authors have gone to some lengths to address the major points and suggestions raised at review. They have added substantial new data in the primary figures and supplementary material. There is also addition of substantial details related to individual cases or examples. The limitation of the manuscript remains the numbers, however that is not something that the authors can reasonably address. Within this limitation, they have satisfactorily addressed each of the points raised by this reviewer.

Reviewer #4 (Remarks to the Author):

The authors present integrative whole genome/transcriptome analyses in 78 cases from the pediatric arm of the Personalized OncoGenomic study from British Columbia in Canada. Although there has been extensive literature on the use of comprehensive genome analyses for children with cancer, I believe this manuscript provides supporting evidence on the value of combined WGS and RNA-seq analyses and is of interest to the field. Overall, the study is well executed and presents some interesting observations especially on DNA-RNA integration and immunogenomic analysis.

I have read the revised version of the manuscript focusing on the points raised by reviewer-3. I believe the authors responded satisfactorily to the reviewer's comments. I have no further comments of my own.

REVIEWERS' COMMENTS

Reviewer #1 (Remarks to the Author):

I am satisfied by the authors responses to the reviewer comments. The clarifications and additions to the manuscript have strengthened it.

Response:

Thank you for your review and approval.

Reviewer #2 (Remarks to the Author):

The authors have gone to some lengths to address the major points and suggestions raised at review. They have added substantial new data in the primary figures and supplementary material. There is also addition of substantial details related to individual cases or examples. The limitation of the manuscript remains the numbers, however that is not something that the authors can reasonably address. Within this limitation, they have satisfactorily addressed each of the points raised by this reviewer.

Response:

We appreciate your recognition of our additions and attempts to address all points raised.

Reviewer #4 (Remarks to the Author):

The authors present integrative whole genome/transcriptome analyses in 78 cases from the pediatric arm of the Personalized OncoGenomic study from British Columbia in Canada. Although there has been extensive literature on the use of comprehensive genome analyses for children with cancer, I believe this manuscript provides supporting evidence on the value of combined WGS and RNA-seq analyses and is of interest to the field. Overall, the study is well executed and presents some interesting observations especially on DNA-RNA integration and immunogenomic analysis.

I have read the revised version of the manuscript focusing on the points raised by reviewer-3. I believe the authors responded satisfactorily to the reviewer's comments. I have no further comments of my own.

Response:

We thank you for your time and additional review, with a focus on comments from Reviewer 3. We appreciate your recognition of the value of this work.